# Dramatic uneven urbanization of large cities throughout the world in recent decades

Liqun Sun [1], Ji Chen [2✉], Qinglan Li [1✉] & Dian Huang [3]

The world has experienced dramatic urbanization in recent decades. However, we still lack information about the characteristics of urbanization in large cities throughout the world. After analyzing 841 large cities with built-up areas (BUAs) of over 100 km² from 2001 to 2018, here we found an uneven distribution of urbanization at different economic levels. On average, large cities in the low-income and lower-middle-income countries had the highest urban population growth, and BUA expansion in the upper-middle-income countries was more than three times that of the high-income countries. Globally, more than 10% of BUAs in 325 large cities showed significant greening ($P < 0.05$) from 2001 to 2018. In particular, China accounted for 32% of greening BUAs in the 841 large cities, where about 108 million people lived. Our quantitative results provide information for future urban sustainable development, especially for rational urbanization of the developing world.

[1] Shenzhen Institutes of Advanced Technology, Chinese Academy of Sciences, 1068 Xueyuan Avenue, Shenzhen 518055, China. [2] Department of Civil Engineering, The University of Hong Kong, Pokfulam Road, Hong Kong, China. [3] National Supercomputing Center in Shenzhen, 9 Duxue Road, Shenzhen 518055, China. ✉email: jichen@hku.hk; ql.li@siat.ac.cn

According to the latest report from the United Nations (UN), the global population in 2018 was 7.6 billion and the urban population was 4.2 billion[1]. By 2050, the global population will reach 9.7 billion, and 68% of the population (i.e., 6.6 billion people) will live in urban areas. Undoubtedly, urban sustainable development is highly related to the future of humanity[2]. However, due to the different levels of socioeconomic development, the process of urbanization is uneven across countries[3]. Usually, in developed countries, urban expansion is adaptable to the population growth. The residents are served by good public services[4] and have access to adequate urban infrastructure, such as water and energy supplies, sanitation[5], education, and green space or parks. For many developing countries, the national economic growth and development are inadequate to meet the needs of a growing urban population[6,7]. In most cases, these cities lack basic infrastructure, and face overcrowding, pollution, and other urban environmental problems[3]. At present, urban sustainable development is an increasing concern worldwide, and it has been enshrined in the 2030 Agenda for the UN's Sustainable Development Goals[8,9]. To achieve the goals, it is essential to track the urbanization progress of cities at different development stages in as detailed a way as possible. A large number of studies have focused on multiple aspects of urbanization from the regional scale[10–14] to the global scale[15–17]. However, a global perspective that clarifies the urbanization characteristics of large cities around the world is still lacking.

Our work was mainly motivated by a desire to address the following questions: (1) how have global large cities developed in recent decades in terms of urban expansion, population growth, and urban greenness change? (2) What are the relationships between the urbanization features and the economic levels? To address both questions, we used remote sensing data and economic data for the period 2001−2018 and the gridded population data from 2000 to 2015 to quantify the urbanization characteristics of 841 large cities throughout the world. A quantitative and comparative study of the urbanization of global large cities has not yet been made available, but it is vital to combat poverty and achieve urban sustainable development in the near future[8,9,18].

## Results

**Built-up area (BUA) expansion**. From 2001 to 2018, the global BUA increased from $7.47 \times 10^5\,\text{km}^2$ to $8.0 \times 10^5\,\text{km}^2$ (Supplementary Fig. 1a), which is equivalent to an increase in the area of 1,130 standard football fields ($7,140\,\text{m}^2$) per day. During this period, the top 10 countries with the greatest BUA expansion (BUAE) were China (47.5% of the global BUA increase), the United States (9%), Nigeria (5.0%), India (3.6%), Indonesia (2.8%), Russia (1.8%), Mexico (1.7%), Malaysia (1.6%), Vietnam (1.5%), and Ghana (1.3%) (Table 1 and Supplementary Fig. 1b). Except for the United States, the other nine countries are developing and emerging countries.

Using the MCD12Q1 product[19], we merged the BUA pixels adjacent to each other into a united urban patch (see Methods). Our analysis focused on the urban patches larger than $100\,\text{km}^2$ (large cities, hereafter) around the world. The advantage of defining large cities in this way is that we were able to include more cities with large BUA areas but relatively smaller populations, such as many low-density cities in the United States and other developed countries. It is worth noting that the large cities studied in this paper included clusters of cities; for example, the Yangtze River Delta (YRD) in China is considered one city, but actually includes four cities: Shanghai, Suzhou, Wuxi, and Changzhou. From 2001 to 2018, the number of the large cities increased from 777 to 841 (Fig. 1a), and the total BUA of the large cities increased from $2.7 \times 10^5\,\text{km}^2$ to $3.08 \times 10^5\,\text{km}^2$, which are

**Table 1 Ranking of the top 10 countries by percentages of the total built-up area (BUA) expansion, BUA in the large cities, and greening BUA in the large cities.**

| Rank | BUA expansion (%) | Total BUA in the large cities (%) | Greening BUA in the large cities (%) |
|---|---|---|---|
| 1 | China (47.5) | United States (27.0) | China (32.0) |
| 2 | United States (9.0) | China (19.0) | United States (19.0) |
| 3 | Nigeria (5.0) | Japan (6.0) | Japan (7.7) |
| 4 | India (3.6) | Brazil (4.0) | Germany (3.9) |
| 5 | Indonesia (2.8) | Germany (2.9) | Brazil (3.0) |
| 6 | Russia (1.8) | India (2.8) | Italy (2.6) |
| 7 | Mexico (1.7) | Canada (2.5) | South Korea (2.3) |
| 8 | Malaysia (1.6) | Australia (2.0) | France (2.2) |
| 9 | Vietnam (1.5) | Mexico (1.9) | Russia (1.4) |
| 10 | Ghana (1.3) | Russia (1.8) | India (1.3) |

BUA expansion (%): the percentage of the total BUA expansion for each country to the total global BUA expansion from 2001 to 2018; total BUA in the large cities (%): the percentage of the BUA of large cities in each country to the total BUA of the 841 large cities; greening BUA in the large cities (%): the percentage of the greening BUA of large cities in each country to the total greening BUA in the 841 large cities.

36.1% and 38.5% of the global BUA in 2001 and 2018, respectively. Once the observations from all the 841 large cities were compared, strong economic patterns emerged from the seemingly diverse BUAE across those large cities (Fig. 1a). According to the country economic classification issued by the World Bank[20], the 841 large cities were divided into four groups: high income (H, gross national income (GNI) more than $12,375 per capita), upper middle income (UM, GNI between $3996 and $12,375 per capita), lower middle income (LM, GNI between $1026 and $3995 per capita), and low income (L, GNI < $1025 per capita) (Supplementary Fig. 2). The large cities are mainly located in high-income (353 large cities) and upper-middle-income (340 large cities) countries, while there are 127 large cities in the lower-middle-income countries and 21 large cities in the low-income countries. The top 10 countries with the largest BUA of large cities are the United States (27.0% of the BUA in 841 large cities), China (19.0%), Japan (6.0%), Brazil (4.0%), Germany (2.9%), India (2.8%), Canada (2.5%), Australia (2.0%), Mexico (1.9%), and Russia (1.8%) (Table 1).

From 2001 to 2018, the average BUAE in high-income countries was only $12.6\,\text{km}^2$ per city, which significantly differs from that of the other three groups ($P < 0.01$; purple lines in Fig. 1b). On average, the BUAE in the upper-middle-income countries is the highest, reaching $38.0\,\text{km}^2$ per city (purple lines in Fig. 1b), which is more than three times that in the high-income countries. In the upper-middle-income countries, there are 61 large cities with BUAE of over $50\,\text{km}^2$, and 51 of them are located in China (magenta points in Fig. 1a). Moscow is the only city in Europe with a large BUAE ($>50\,\text{km}^2$). In contrast, most of the cities in Europe and South America have BUAE of $<50\,\text{km}^2$ (green points in Fig. 1a), and urban expansions in some cities are almost stagnant ($<1\,\text{km}^2$) (blue points in Fig. 1a).

**Urban population growth**. Urban population growth is one of the driving forces of urban expansion. Using the global population grid data sets[21] in 2000, 2005, 2010, and 2015 (see Methods), we found that 80% of the large cities (676 out of 841) experienced population growth. Among them, 22 large cities experienced population growth of more than 2 million (magenta points labeled with the cities' names in Fig. 1c). These large cities are mainly located in Asia (15 large cities) and Africa (5 large cities), and there is 1 in South America (Sao Paulo) and 1 in Europe

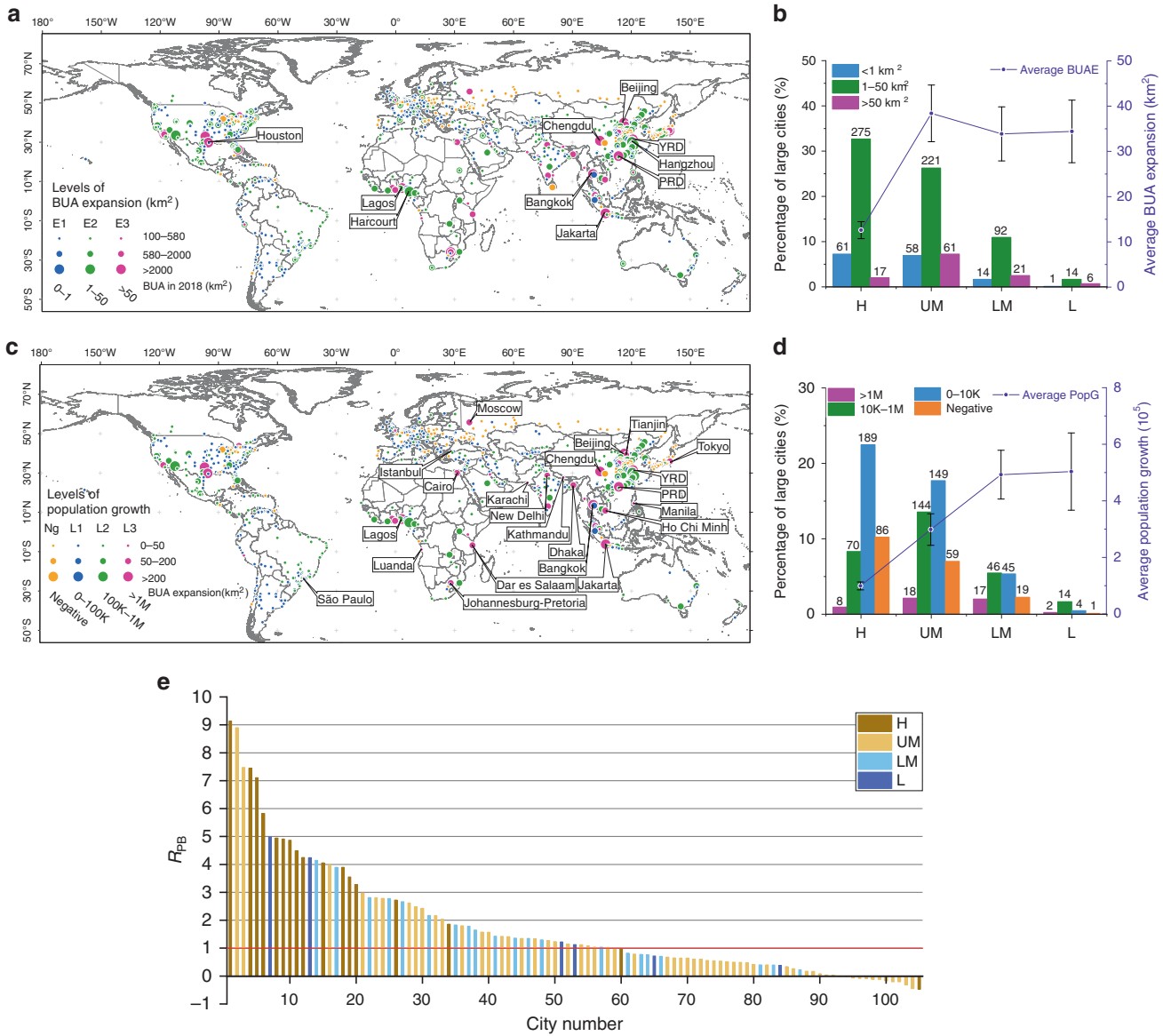

**Fig. 1 Distribution of large cities with the built-up area (BUA) expansion and population growth. a** Distribution of the 841 large cities at different levels of the BUA expansion (BUAE) from 2001 to 2018. E1: $0 \leq$ BUAE $\leq 1\,km^2$, blue points; E2: $1\,km^2 <$ BUAE $\leq 50\,km^2$, green points; E3: BUAE $> 50\,km^2$, magenta points. The top 10 cities with BUAE are labeled with their names. YRD Yangtze River Delta, PRD Pearl River Delta. **b** Bars are the percentages of the total 841 large cities for the large cities at the different levels of BUAE and 4 economic levels. Purple lines are the average BUAE values of the large cities at different economic levels. **c** Distribution of 841 large cities at different levels of population growth (PopG). Ng: negative population growth, orange points; L1: $0 <$ PopG $\leq 100K$, blue points; L2: $100K <$ PopG $\leq 1M$, green points; L3: PopG $> 1M$, magenta points. The 22 large cities with a PopG of more than 2 million are labeled with their names. **d** Bars are the percentages of all the large cities for PopG at four different economic levels. Purple lines are the values of average population growth of large cities at different economic levels. All large cities are classified according to the World Bank's estimates of 2018 gross national income (GNI) per capita, H high income, UM upper middle income, LM lower middle income, and L low income; see Methods for details. **e** The ratio ($R_{PB}$) of the urban population growth rate to the BUAE rate for the 105 large cities with BUAE of over $50\,km^2$ from 2001 to 2018 (magenta points in (**a**)). For each city, the urban population growth rate was calculated using the data for the period from 2000 to 2015. In (**b**, **d**), the error bars show SEM (standard error of the mean) for each economic level.

(Moscow) (Fig. 1c). The average population growth rates of the large cities in the low-income and lower-middle countries from 2000 to 2015 are almost the same, $\sim 5.0 \times 10^5$ per city, which is more than five times that in the high-income countries ($1.0 \times 10^5$ per city) and 1.7 times that in the upper-middle-income countries ($3.0 \times 10^5$ per city) (Fig. 1d). We calculated the ratio ($R_{PB}$) of the urban population growth rate (UPGr) to the BUAE rate (BUr) for the 841 large cities (see Methods), and found that the $R_{PB}$ is larger than 1 in 543 large cities (64.5%). In particular, for the 105 large cities with BUAE of over $50\,km^2$, the UPGr in 55% of the large

cities (58 out of 105) is faster than the urban expansion rate (Fig. 1e), even though these large cities have already experienced a significant BUAE. This result indicates that in most parts of the world, without considering the vertical urban growth through the construction of high-rise apartments and condominiums, urban expansion has lagged behind the population growth since the advent of the new century. This is exactly opposite to the trend in the urbanization before 2000 reported by a previous study, which found that urban land was expanding faster than the urban population growth[22].

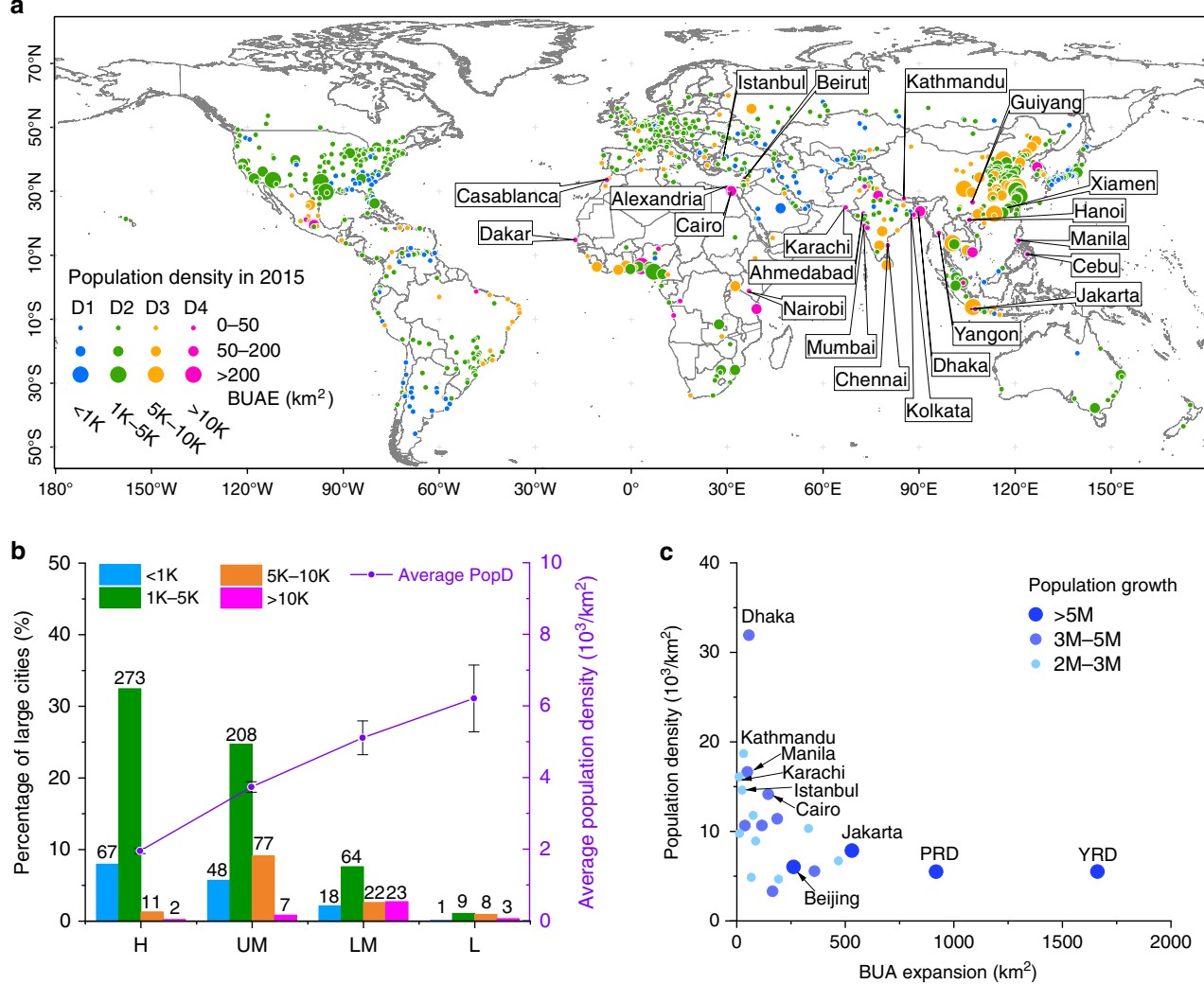

**Fig. 2 Distribution of large cities with the population density. a** Distribution of the 841 large cities at four different levels of the population density (PopD, unit: 1/km$^2$) in 2015. D1: 0 < PopD ≤ 1K, blue points; D2: 1K < PopD ≤ 5K, green points; D3: 5K < PopD ≤ 10K, orange points; D4: PopD > 10K, magenta points. **b** Bars are the percentages of the 841 large cities for four PopD levels in 2015 at four economic levels. Error bars show SEM for each economic level. **c** Relationship between PopD in 2015 and the BUA expansion from 2001 to 2018 of the 22 large cities with population growth larger than 2M. YRD Yangtze River Delta, PRD Pearl River Delta. The size of the dots refers to the population growth at different levels.

In 2015, the 20 cities with the highest population density were located in Asia (15 large cities) and Africa (5 large cities) (Fig. 2a). The average population density of the large cities in the low-income countries was $6.2 \times 10^3$ per km$^2$, which is ~3.2 times that in the high-income countries ($2.0 \times 10^3$ per km$^2$) (purple lines in Fig. 2b). Among these large cities, Dhaka, Kathmandu, Manila, Karachi, Istanbul, and Cairo are the top 6 large cities with the highest population density (Fig. 2c). These cities experienced rapid population growth (>2 million) from 2000 to 2015 (Fig. 2c), and limited urban expansion from 2001 to 2018 (<200 km$^2$, Fig. 2c). In contrast, during the same period, there were only four large cities experiencing population growth of more than 5 million, and they are the YRD, Pearl River Delta (PRD), Beijing, and Jakarta. Due to rapid urban expansion, the population densities of these four cities were relatively low (Fig. 2c).

**Urban greenness change**. To obtain a rational perspective of global urban greenness change, we assumed that if the greenness of a BUA pixel increased, some new parks, green spaces, and/or green roofs might have been developed in the pixel, or, at a

minimum, the street vegetation in the pixel experienced growth. To avoid interference from the vegetation phenology, we chose the annual maximum greenness of the vegetation to represent the best state of urban greening.

Using the enhanced vegetation index (EVI)[23,24] as a greenness indicator and BUA of 2018 as the urban extent, we applied the Mann–Kendall method to evaluate the trend of the annual maximum EVI (EVI$_{max}$) for each urban pixel of the 841 large cities from 2001 to 2018 (see Methods). Then, we calculated the ratio ($R_{greening}$) of the area of the pixels with a significant increasing EVI$_{max}$ trend ($P < 0.05$) to the BUA in 2018 of the corresponding large city. Using this method, we evaluated urban greening change since 2001. Most of the pixels with significant greening trends were located in stable BUAs (i.e., pixels that were BUAs throughout the study period, 2001–2018; see Methods). Finally, all 841 large cities were divided into three levels according to the values of $R_{greening}$ (R1–R3, Fig. 3a). There are 325 large cities where more than 10% of the urban pixels (i.e., $R_{greening} >$ 0.1) indicated significant greening from 2001 to 2018 (magenta points in Fig. 3a). These large cities are predominantly distributed in East Asia, Europe, and North America, and sparsely distributed

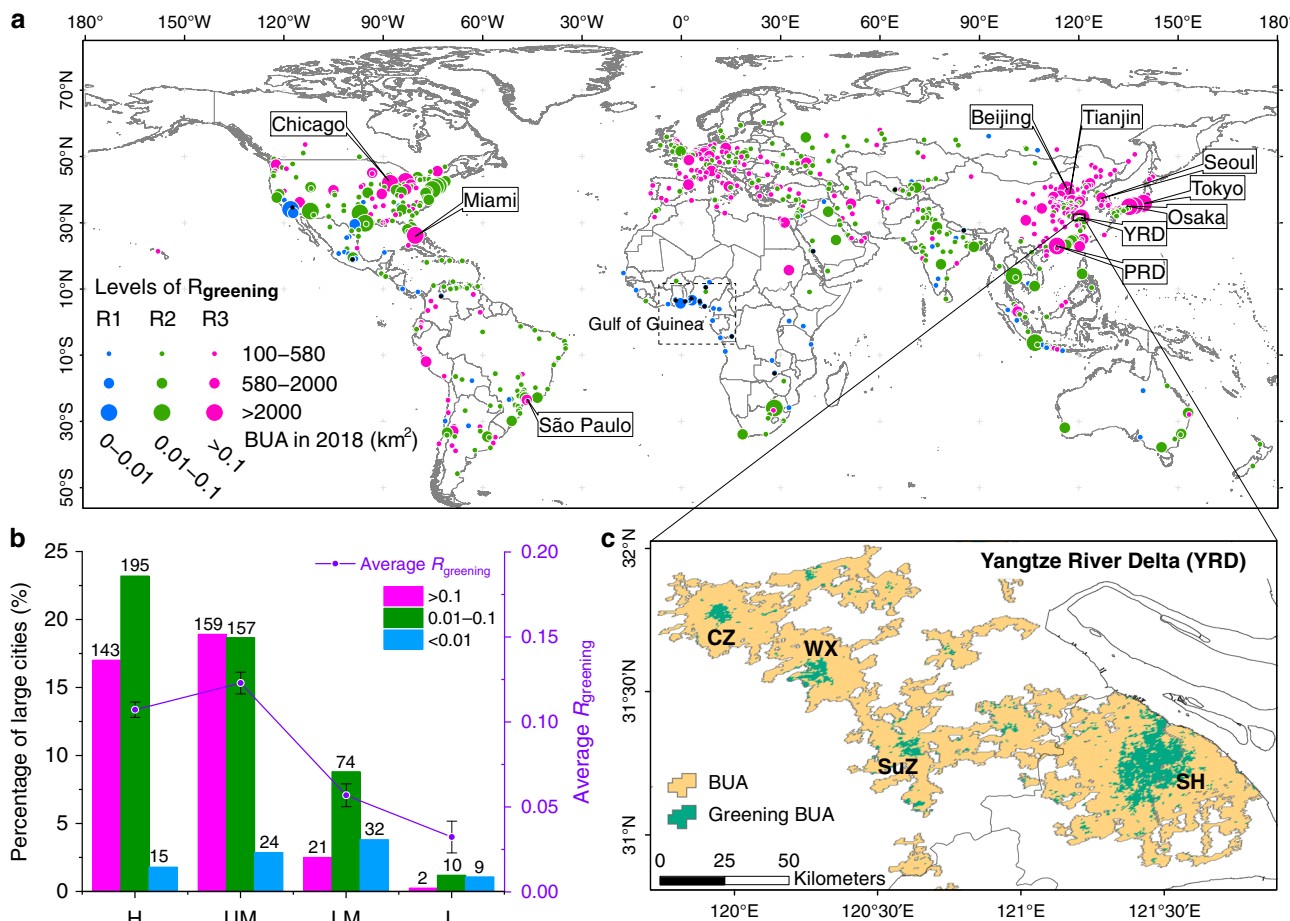

**Fig. 3 Distribution of large cities with greening built-up area (BUA). a** Distribution of the 841 large cities at different levels of greening BUA ($R_{greening}$, the ratio of the BUA pixels with significant greening trend to the total BUA for a large city). R1: $R_{greening} \leq 0.01$, blue points; R2: $0.01 < R_{greening} \leq 0.1$, green points; R3: $R_{greening} > 0.1$, magenta points. The top 10 greening BUA of large cities are labeled with their names. The large cities with no pixel greening are overlapped with black dots (i.e., the cities in the dash box of the Gulf of Guinea in Africa). **b** Percentages of the 841 large cities for three different $R_{greening}$ levels and four economic levels. Purple lines are the average values of $R_{greening}$ at different economic levels. Error bars show SEM for each economic level. **c** An example of greening BUA in YRD of China. YRD Yangtze River Delta. YRD mainly includes four cities: Shanghai (SH), Suzhou (SuZ), Wuxi (WX), and Changzhou (CZ).

in South America, Africa and Oceania. In particular, China, the United States, and Japan are the top 3 countries contributing to the greening from 2001 to 2018, accounting for 32%, 19%, and 7.7% of the total greening BUA in the 841 large cities, respectively (Table 1). Moreover, there are 159 large cities with $R_{greening}$ larger than 0.1 in the upper-middle-income countries, and 143 large cities in high-income countries (magenta bars in Fig. 3b). Among the four economic classifications, the average value of the $R_{greening}$ is the largest in the upper-middle-income countries, reaching 0.12 per city, followed by the high-income countries with 0.1 per city. The average value of $R_{greening}$ in the low-income countries is the smallest, ~0.03 per city (purple lines of Fig. 3b).

From the spatial pattern of the greening pixels in 325 large cities ($R_{greening} > 0.1$), we found that many greening areas of large cities have a shape like a "fried egg" where the yolk-shaped downtown area has been turning green significantly ($P < 0.05$); the "egg white" area refers to the area of browning or no significant greening outside the yolk-shaped area, and these areas are usually the suburbs. As one of the ten large cities with the largest greening BUA from 2001 to 2018, the YRD is a typical example of the "fried egg" with a cluster of the yolk-shaped areas of greening BUA (Fig. 3c). The other nine large cities with the largest greening BUA are PRD, Tokyo, Miami, Beijing, Chicago, Seoul, Tianjin, Sao Paulo, and Osaka (Fig. 3a, Supplementary

Fig. 3, and Supplementary Table 1). The yolk-shaped greening BUA is also obvious in Beijing, Chicago, Seoul, Tianjin, Sao Paulo, and Osaka (Supplementary Fig. 3d–i). These yolk-shaped greening areas are usually located in the downtown of the cities. Normally, the main characteristics of these areas include a high building density and high population density[25]. In these areas, the majority of the urban construction would have already been completed before 2001, so any newly established parks or green spaces, even the growth of street vegetation, would increase the greenness of the BUA pixels year by year. Notably, although the PRD has the largest area of greening BUA (Supplementary Table 1), there are many hills and mountains in that region (Supplementary Fig. 4a); thus, the greening area distributed in the PRD is rather diffuse compared to that in the YRD (Supplementary Fig. 4b).

**Population in greening BUA.** According to the latest gridded population data in 2015, there are ~1.16 billion urban residents in the 841 large cities, and 192 million urban population are living in the BUA pixels with significant greening trends (Fig. 4a). The average population living in BUA with significant greening trends in the upper-middle-income countries is the largest, ~$4.0 \times 10^5$ per city, followed by the high-income countries with a value of

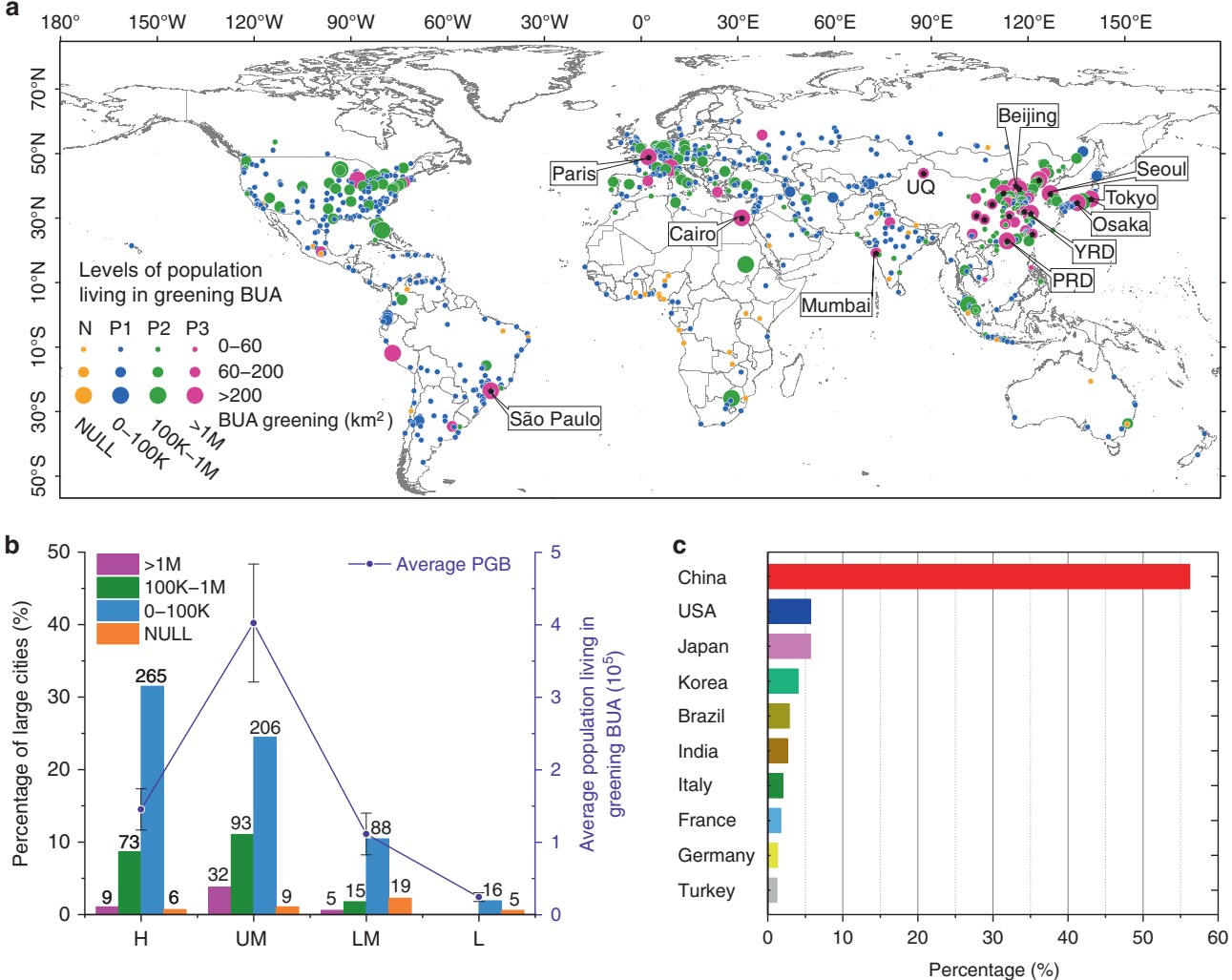

**Fig. 4 Distribution of the population living in greening built-up area (BUA). a** Distribution of the 841 large cities at different levels of population living in greening BUA (noted as PGB). N: No PGB; P1: 0 < PGB ≤ 100K, blue points; P2: 100K < PGB ≤ 1M, green points; P3: PGB > 1M, magenta points. The top 20 PBG large cities are marked with black dots, and 10 of them are labeled with their names (see Supplementary Table 2 for details). YRD Yangtze River Delta, PRD Pearl River Delta, UQ Urumqi. **b** Bars are the percentages of the total 841 large cities for PGB at four economic levels. Purple lines are the average PGB at four economic levels. Error bars show SEM for each economic level. **c** The top 10 countries with largest percentages of the total PGB in the 841 large cities.

$1.5 \times 10^5$ per city (Fig. 3b). In contrast, the average numbers for the urban population living in the greening BUA in the lower-middle-income and low-income countries are only $1.1 \times 10^5$ per city and $0.24 \times 10^5$ per city, respectively (Fig. 4b). As the most populous country, China has 150 large cities, and there are about 108 million urban population living in greening BUAs. This accounts for 56.2% of the total population in greening BUAs among the 841 large cities (Fig. 4c). The United States has 151 large cities, with 10.98 million people living in greening BUA (Fig. 4c). As an emerging economy and the second most populous country in the world, India has 34 large cities, in which 5.2 million people live in greening BUA (Fig. 4c). The 20 cities with the largest population living in greening BUA (see Supplementary Table 2) are marked with black points in Fig. 4a. Among them, Urumqi (UQ, in Fig. 4a), the capital of the Xinjiang Autonomous Region in the semi-arid region of north-western China, has the highest value of $R_{\text{greening}}$ (0.53) (Supplementary Table 2).

In fact, the greenness of many cities in upper-middle-income countries increased significantly from 2001 to 2018, which was due to the relatively low greenness of these cities before 2001. In contrast, in the high-income countries, there were quite

possibly already many parks and green spaces in the cities[4,26,27] before 2001 (Supplementary Fig. 5a). Even though the urban greenness of upper-middle-income countries increased significantly in recent decades, cities in the high-income countries in 2018 were still "greener" compared to those in developing countries. To test this understanding, we calculated the average $EVI_{\text{max}}$ values of all the large cities ($EVI_{\text{city}}$) in 2018, and then ranked them into four quarters (Q1–Q4 in Supplementary Fig. 5b). Notably, due to the semi-arid or arid climate, the urban greenness of large cities located in the southwest of the United States and Australia is usually low (magenta points of Q1 in Supplementary Fig. 5a). However, our results reveal that the economic impacts on urban greenness are significant (Supplementary Fig. 5c). The average $EVI_{\text{city}}$ in high-income countries is 0.37, which is the highest among the four economic classifications. The average $EVI_{\text{city}}$ in the low-income countries is 0.28. Furthermore, according to the gridded population data in 2015, only 12% (135 million) of the large cities' total population lived in cities in Q4; moreover, over 71% (157 out of 211) of these large cities in Q4 were located in high-income countries (Supplementary Fig. 5d). By contrast, ~69% of the

large cities' total populations lived in cities in Q1 and Q2 (Supplementary Fig. 5d).

## Discussion

Using the results presented above, we identified the urbanization characteristics for large cities in terms of urban expansion, population growth, and greening BUA changes. We observed that global urbanization in the countries with different income classifications in recent decades exhibited a dramatic unevenness. Below, we highlight three urbanization features obtained in this study.

The first is the unevenness between the BUAE and urban population growth. From 2001 to 2018, the urban expansion and urban population growth in the high-income countries were the lowest (12.6 km$^2$ per city; $1.2 \times 10^5$ per city) among the four classifications (Fig. 1b, d). Conversely, the upper-middle-income countries had the largest urban expansion (38.0 km$^2$ per city) (Fig. 1b), while the low-income countries had the highest urban population growth ($5.0 \times 10^5$ per city) (Fig. 1d). From the ratio of the UPGr to the BUA growth rate, we found the urban population increased faster than the urban expansion in most parts of the world in recent decades. This result could be attributed to the fastest-growing urban population since the 1950s, which has completely changed the urbanization state from mid-1980s to 2000 reported by a previous study[22]. Moreover, due to the different growth rates of the urban population and urban area, the unevenness of the population pressure in the high-income countries ($2.0 \times 10^3$ per km$^2$) and low-income countries ($6.2 \times 10^3$ per km$^2$) is also significantly different ($P < 0.01$) (Fig. 2b). This result indicates that compared with the faster-growing urban population (Fig. 1d), even though the BUAE of the low-income countries is relatively fast (Fig. 1b), the urban expansion still lags far behind the urban population growth (Fig. 2b). Hence, the cities with a rapid population growth and relatively slow BUAE in the lower-middle-income and low-income countries could undergo serious urban problems, such as slums and crowding[15,28]. In addition, due to the fast urban expansion, the urbanization in the upper-middle-income countries may meet the needs of their population growth.

The second feature is unevenness between rapid urban population growth and slow urban greening increase. We used the annual maximum greenness (EVI$_{max}$) to represent the best state of urban greening. According to the average EVI$_{max}$ for each city (EVI$_{city}$) in 2018, we found that one quarter of the greenness with the highest range of EVI$_{city}$, Q4, only encompassed 12% of the total population in the large cities (Supplementary Fig. 5d). In contrast, ~69% of the total population lived in the areas with a lower greenness (Q1 and Q2). However, with the economic development of the upper-middle-income countries in recent decades, urban greening has increased significantly, where the average $R_{greening}$ is 0.12 (Fig. 3b). In the high-income countries, due to "greener" urbanization (Supplementary Fig. 5a) before 2001, the greening increase of these cities is not the largest (Fig. 3b). Nevertheless, in the low-income countries, not only is the greenness of cities the lowest (EVI$_{city}$ = 0.28; Supplementary Fig. 5c) but the ratio of greening BUAs from 2001 to 2018 is also the smallest ($R_{greening}$ = 0.04; Fig. 3b). These results indicate that on the way to equitable and sustainable urban development[29], many large cities in the upper-middle income countries initially made substantial progress, but there is still a long way to go for the large cities in lower-middle-income and low-income countries. In addition, as the world's largest source of emissions[30], large cities with a significant greening trend may also benefit the health of local urban residents[31,32], and, to a certain extent, mitigate the impact of global climate change[33].

The third feature is rapid urbanization of the large cities in China. From 2000 to 2015, the urban population growth of China was the largest in the world[1]. The urban expansion from 2001 to 2018 was also the largest, accounting for 47.5% of the total urban expansion in the world (Supplementary Fig. 1b). Due to the rapid economic growth in the study period[34], China invested a large amount of resources into infrastructure construction for advancing the urban living environment[35]. Among the 325 large cities in the world with significant urban greening increase ($R_{greening} >$ 0.1) (magenta points in Fig. 3a), 101 of them are located in China, and most of them have remarkable yolk-shaped greening areas (Fig. 3a). These greening urban areas account for 32% of the total greening BUAs in the 841 large cities, where 108 million people were living (Fig. 4c). The greening of cities would have resulted from more urban parks or green spaces, which are likely to have multiple benefits for the urban environment and human health/ well-being, including improving cardiovascular conditions[31,36], and mental health[32] and reducing crime and violence[37].

Overall, we observed that urbanization in the large cities has progressed rapidly in recent decades; moreover, our results indicate there are some economic-related global urbanization characteristics. However, due to the scarcity of the economic data (income levels)[20], the interpretation of the results still has certain limitations. For example, with a large economic aggregate and large land area, China has a significant difference in the economic development in the eastern and western regions. In the YRD and PRD in eastern China, the gross domestic product (GDP) per capita in 2018 (see Methods) has reached the level of the high-income countries. Thus, we should be cautious in elucidating the impacts of country income levels on urbanization. In addition, in order to provide a global estimate of urbanization, we used 100 km$^2$ as a criterion to screen large cities throughout the world. Consequently, only dozens of the cities in Africa and India were selected, which may not represent the reality of the urbanization process in those countries. In a future study, it will be necessary to adjust the criterion for large cities to get a more comprehensive understanding of the urbanization characteristics in those countries or regions.

The systematic monitoring of the global urbanization process, especially in the low-income countries where the data are normally scarce, is essential to achieving the UN's Sustainable Development Goals[8,9]. Therefore, our research using remote sensing data associated with limited demographic and economic data indicates a viable way of synoptically and repeatedly monitoring urbanization process worldwide. Our quantitative results, combined with previous policy studies[3,15,38] and case studies[14,39,40], make it possible to assess the unevenness of the urbanization from more perspectives, such as the urban heat island effect[41], water security[42], and structure[25]. For the majority of the developing countries, understanding the dramatic uneven urbanization of the past several decades can provide a scientific reference to improve their urban governance[43], thereby achieving a virtuous circle among urban expansion, population growth, and urban greening changes. This is vital for achieving urban sustainable development.

## Methods

**MODIS land cover type product.** The urban and built-up pixels used in this study were extracted from the International Geosphere-Biosphere Programme classification types provided by the collection 6.0 MODIS yearly product known as MCD12Q1. The spatial resolution of the land cover product was 500 m and had a reported 73.6% overall land cover classification accuracy and relatively high accuracy for urban areas[19]. Any pixel classified as urban and BUA had at least a 30% impervious surface area, including building materials, asphalt, and parking lots. We used the tool "Raster to Polygon" in ArcGIS software to extract the BUA polygon from the raster layer. In this process, only the edge-adjacent (four connectivity) pixels were merged into a united urban patch.

In this study, there were three levels of BUAE (Fig. 1a): E1 refers to very small or no urban expansion (≤1 km$^2$), E2 is less than half of the minimum BUA of the

large city (1 km$^2$ < BUAE ≤ 50 km$^2$), and E3 is larger than half of the minimum BUA of the large city (>50 km$^2$). The reason for using 50 km$^2$ as the threshold was that when excluding 12 large cities with BUAE of more than 250 km$^2$, we found that the value of the average urban expansion of the remaining 829 large cities, adding 1 standard deviation to those expansions, was close to 50 km$^2$.

In addition, there were three levels of BUA (Fig. 1a). The BUA of the first level was between 100 and 580 km$^2$, the second level was between 580 and 2000 km$^2$, and the third level was larger than 2000 km$^2$. For BUAs in 2018, more than 19 large cities had areas over 2000 km$^2$. Excluding those 19 cities, we found that the value of the average BUA of the remaining 822 large cities, adding 1 standard deviation to those areas, was close to 580 km$^2$. This is why we divided the 841 large cities into three different sizes (i.e., the point sizes in Fig. 1).

**EVI data and EVI$_{max}$ trends**. The vegetation index data sets from the MOD13A1 version 6 product were used in this study to match the spatial resolution of the land cover data (500 m). These data sets include two primary vegetation layers: the normalized difference vegetation index (NDVI) and the EVI. Due to the greater sensitivity of the EVI relative to that of the NDVI in the high-biomass regions, our results were based on the use of the EVI. For each pixel, the annual maximum EVI (EVI$_{max}$) was generated from all the acquisitions over a 16-day period.

We used a Mann–Kendall test, which is a nonparametric test, to detect monotonic trends in the time-series data and evaluate the trends of EVI$_{max}$ from 2001 to 2018. For a comprehensive and systematic analysis of global urban greenness change, we used the BUA in 2018 to detect urban greening change. Then, if a pixel changed to an urban area in 2002 or later, the greenness change in the pixel was calculated. Notably, if the expended BUA was developed from cropland and natural vegetation land, the greenness of the expended BUA would possibly decrease. In fact, we calculated the BUA where greenness showed a significant increasing trend ($P < 0.05$). Using this method, we could exclude those pixels that had changed from rural vegetation to BUA (urban expansion).

In this study, the spatial resolution of the MOD13A1 data was 500 m; thus, we wrote a Python program to calculate the Mann–Kendall trend of EVI$_{max}$ from 2001 to 2018 for each pixel in a raster file. Then, the Geospatial Data Abstraction Library (https://gdal.org) was used to mosaic all the raster files (212 files) of the trends into a global raster map of the EVI$_{max}$ trend. Finally, we extracted the EVI$_{max}$ trend for each urban pixel within the urban extent (BUA in 2018) and calculated the ratio ($R_{greening}$) of the area of the pixels with a significant increasing EVI$_{max}$ trend ($P < 0.05$) to the entire BUA of the corresponding urban extent.

**Global country administrative area**. The global country administrative areas were downloaded from the Database of Global Administrative Areas (GADM). The GADM data consisted high-resolution shapefiles at all administrative levels, such as at the country, state, and provincial levels. We used the latest version (v 3.4) in this study.

**Population and economic data**. The country population data were obtained from *World Urbanization Prospects: The 2018 Revision*. The 1 km resolution Gridded Population of the World collection (fourth version; GPWv4), which was adjusted to the national-level population counts estimated by the UN, was obtained from the Socioeconomic Data and Applications Center. The World Bank classifies economies as low income (<$1025), lower middle income ($1026–$3995), upper middle income ($3996–$12,375), or high income (more than $12,375) based on the GNI per capita in 2018. The GDP data of YRD and PRD in 2018 were obtained from the Statistics Bureau of Jiangsu Province and Guangdong Province, respectively.

In this study, we calculated the ratio ($R_{PB}$) of the UPGr to the BUr for the 841 large cities. For each large city, the UPGr and BUr were defined as follows:

$$UPGr = \left(Pop_{2015} - Pop_{2000}\right)/Pop_{2000},$$

$$BUr = \left(BUA_{2018} - BUA_{2001}\right)/BUA_{2001},$$

where Pop$_{2015}$ and Pop$_{2000}$ are the urban populations of a large city in 2015 and 2000, respectively. BUA$_{2018}$ and BUA$_{2001}$ are the BUAs of the large city in 2018 and 2001, respectively.

**Reporting summary**. Further information on research design is available in the Nature Research Reporting Summary linked to this article.

## Data availability

The MODIS land cover type products are available from https://e4ftl01.cr.usgs.gov/MOTA/MCD12Q1.006/. The MODIS EVI data sets are available from https://e4ftl01.cr.usgs.gov/MOLT/MOD13A1.006/. The global country administrative areas are available from the Database of Global Administrative Areas (GADM) (https://www.gadm.org/). The country population data were obtained from *World Urbanization Prospects: The 2018 Revision* (https://population.un.org/wup/). The 1 km resolution Gridded Population of the World (GPW) collection (fourth version; GPWv4) is available from the Socioeconomic Data and Applications Center (https://sedac.ciesin.columbia.edu/data/collection/gpw-v4/united-nations-adjusted). The World Bank classifies economies based

on the gross national income (GNI) per capita in 2018 (https://blogs.worldbank.org/opendata/new-country-classifications-income-level-2019-2020). The GDP data of YRD in 2018 were downloaded from the Statistics Bureau of Jiangsu Province (http://tj.jiangsu.gov.cn), and the GDP data of PRD in 2018 were downloaded from the Statistics Bureau of Guangdong Province (http://stats.gd.gov.cn/).

## Code availability

All computer code used in the data analysis is available from the corresponding author upon reasonable request.

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

## Acknowledgements

This study was supported by the Science and Technology Department of Guangdong Province with Grant of 2019B111101002, the Innovation of Science and Technology Commission of Shenzhen Municipality Ministry with Grants of JCYJ20170413164957461 and GGFW2017073114031767, and National Natural Science Foundation of China (Grant No. 91747205).

## Author contributions

L.S. and J.C. designed the research; L.S., Q.L., and D.H. conducted data analysis and calculation; L.S. and J.C. wrote the paper; all the authors contributed to the interpretation of the results.

## Competing interests

The authors declare no competing interests.
