## [Peer Review File · Nature Communications]

Reviewers' comments:

Reviewer #1 (Remarks to the Author):

I have reviewed NCOMMS-19-42313, "How have global large cities grown in recent decades". I regret that there are several major problems with this paper and I do not think it is publishable as written. Here are my my main concerns:

1) The purpose of the paper is unclear. The clearest articulation of what the paper is about is found in the title, but after the title the hypotheses, research problems and or purpose for conducting this study are never articulated. The paper becomes a description of how cities have grown and how their greenness has changed. There seems to be a particular focus on comparing China with the USA. I recognize China is the world's largest country with one the fastest growing urban populations, but the emphasis was somewhat strange and seemed to reveal a biased point of view of the authors, that came across as propoganda at times (e.g. see the discussion and conclusions on line 160-173).

2) The writing in the paper was poor. The abstract and introduction had multiple errors in every paragraph, making the paper very hard to read. The quality of the writing improved through the paper, but the paper needs to be rewritten and restructured to be clearer, more logical and objective in its scholarship.

3) The conclusion of the authors were often subjective, lacking the objective rigor of scholarship I expect from scientific articles. The authors seem to be making subjective value judgements about the effects of urban growth in some countries, that seem completely unrelated and unsubstantiated by their data. This starts in the introduction on lines 96-98, and continues in the conclusions (e.g. line 169; lines 171-173; 177-179, line 181-182; line 189).

In conclusion, I think the data is interesting, but the reason of the study is unclear, the writing is poor and the conclusions are biased an go beyond the data.

I regret the paper is not publishable in its current form in a journal.

Reviewer #2 (Remarks to the Author):

Overall, this is a good paper which quantifies the uneven urbanization from individual city to global scale. The authors analyzed urbanization characteristics in terms of urban expansion, population growth, built-up area greening and its beneficiaries, and found these characteristics show strong economic-related or spatial patterns. The paper is well-written, clearly organized, and the analysis is well conducted. Hence, I recommend it for publication after addressing the comments suggested below that authors may consider for improving their manuscript.

1. The concept/term of urbanization is ambiguous in this paper. It is sometimes used to refer to 'urban expansion' (e.g., Section 'Rapid urbanization process'), which is the physical growth of built-up areas. While in some other contexts, the term refers to the general process including urban expansion, population growth and urban environment improvement, e.g., in Lines 160-161. So it would better to clarify and unify the meaning of the term 'urbanization'.

2. The economic data used in the analysis is based on economic classification at the country level. But the economic level in one single country can be very uneven. Take China as an example. Regions along China's eastern coast are far better developed than other parts of China. Therefore, it's not quite appropriate to group cities like Urumqi in the same economic level with cities/urban agglomerations such as Beijing, PRD and YRD.

I understand that maybe using country-level data is the only feasible way for this analysis at the global scale. So if there are no other suitable economic level datasets, this limitation should at

least be mentioned in the discussion section.

For the case studies of PRD and YRD, authors could consider using the economic data of these regions alone instead of the number for the whole country, to quantify the relationship between levels of socio-economic and BUA, population growth, urban greening, etc.

3. The term 'Large cities' is commonly defined based on population. In this paper, large cities are urban patches larger than 100 km² extracted from satellite images. Could the authors clarify if this definition will omit cities with a large population or high population density but relatively small areas?

4. Fig. 1 and Supplementary Fig. 5: It would be better to consider combining BUA and population, i.e., calculating BUA population density, in Fig. 1 and Supplementary Figure 5. In this way, the situation of urban land expansion lagging behind the needs of the growing urban population (overcrowding) in lower-middle-income and low-income countries can be better presented.

5. Figures 1b, 1d, 2b and 3b: I suggest that the authors conduct a statistical test to see if the average BUA, average PopI, average REVI and the average population living in BUA greening at the four economic levels are significantly different.

6. Lines 130-139: The "fried egg" shape is very interesting. But it would be better to briefly explain what the "fried egg" shape indicates and why this shape matters. For example, what is the difference between the 'yolk' areas and 'egg white' areas in terms of urban density, population density or time when the urban land appears? Why it's shaped like that?

7. Section 'Beneficiaries from urban greening': Greening areas in this context are not necessarily greener than other areas. Just as said in Line 189, some areas are "green enough". So besides beneficiaries of urban greening, I suggest analyzing beneficiaries of urban greenness (EVI_{max}) as well, instead of relating urban greenness to BUA in Supplementary Fig. 6.

8. Some other minor questions:

1) Line 9: Please check the grammar. 'Global' is an adjective.

2) Lines 10-11: In the sentence 'we found lower-middle-income and low-income countries respond to the high population pressure and environmental problems', the word 'respond' seems unclear.

3) Line 51: Please indicate if 'adjacency' is determined based on 4-connectivity or 8-connectivity.

4) Section 'MODIS land cover type product': The classification accuracy of the MODIS land cover product could be mentioned in order to show the reliability of using this data to extract large cities.

5) Lines 222-223: It is not clear how NDVI datasets were used in the study. If it is important, it would be better to give a little more explanation. Otherwise, I suggest deleting the sentence 'the NDVI datasets are used for the comparative analyses'.

6) The word 'frequency' in all of the figures seems a little unsuitable. 'Percentage' may be the better word. It would also be better if the decimals in Table 1 could be presented as percentages.

7) Fig. 3b: The title of the purple y-axis should be 'Average population living in GBUA (thousand)'.

I hope the authors will find my comments useful.

Response to Reviewers

We much appreciate the valuable comments and suggestions from both the reviewers. These comments and suggestions have encouraged us to view our work with much greater insight than before. According to these comments, we have revised the manuscript thoroughly. The detailed responses to these comments are given below.

Response to Reviewer #1:

I have reviewed NCOMMS-19-42313, "How have global large cities grown in recent decades". I regret that there are several major problems with this paper and I do not think it is publishable as written. Here are my main concerns:

1) The purpose of the paper is unclear. The clearest articulation of what the paper is about is found in the title, but after the title the hypotheses, research problems and or purpose for conducting this study are never articulated. The paper becomes a description of how cities have grown and how their greenness has changed. There seems to be a particular focus on comparing China with the USA. I recognize China is the world's largest country with one the fastest growing urban populations, but the emphasis was somewhat strange and seemed to reveal a biased point of view of the authors, that came across as propaganda at times (e.g. see the discussion and conclusions on line 160-173).

Response: Thanks for the valuable comments. We have revised the paper thoroughly and clarified the research problems and the purpose of the paper (Please see the text on lines 44-51 of the revised manuscript). To highlight the findings of the study, we also amended the paper title as "Dramatic uneven urbanization development of global large cities in recent decades".

Urbanization is an unavoidable socioeconomic progress; specifically, due to rapid population growth and economic development since the middle twentieth century, the world has experienced dramatic urbanization. However, because of the urban data availability, a systematic understanding of global urbanization has not been available yet. Undoubtedly, such an understanding is valuable for disclosure of likely experiences and lessons of urbanization in different parts of the world. Using the state-of-the-art urban data, including built-up area, population and land vegetation index, the paper aims to reveal the features of urbanization development of global large cities in the recent decades (from 2001 to 2018). Therefore, as indicated in the text on lines 44-47, three research questions which motivated the study are "(i) How has global urbanization developed in recent decades? (ii) What is the relationship between the economy and urban development? (iii) Has the urban environment improved? If so, how many people have benefited?".

The text on lines 160-173 in the original submission has been removed. Specifically, we have removed the description which may cause a biased point of view. Nevertheless, as a salient urbanization development from 2001 to 2018, the urban expansion in China from 2001 to 2018 is the largest, accounting for 47.5% of the total urban expansion in the world. Therefore, the paper includes the texts on lines 241-252 to provide the salient urbanization feature in China.

2) The writing in the paper was poor. The abstract and introduction had multiple errors in every paragraph, making the paper very hard to read. The quality of the writing improved through the paper, but the paper needs to be rewritten and restructured to be clearer, more logical and objective in its scholarship.

Response: We have revised the paper thoroughly. Also, this revised manuscript has been edited by LetPub Editing Service.

3) The conclusion of the authors was often subjective, lacking the objective rigor of scholarship I expect from scientific articles. The authors seem to be making subjective value judgements about the effects of urban growth in some countries, that seem completely unrelated and unsubstantiated by their data. This starts in the introduction on lines 96-98, and continues in the conclusions (e.g. line 169; lines 171-173; 177-179, line 181-182; line 189).

Response: We rewrote the section of Discussion and conclusions (Lines 198-278). We removed those sentences which might cause subjective conclusions. In this revised manuscript, all the results were induced from data analysis with objective rigor of scholarship.

In conclusion, I think the data is interesting, but the reason of the study is unclear, the writing is poor and the conclusions are biased and go beyond the data.

I regret the paper is not publishable in its current form in a journal.

Response: We much appreciate the reviewer's constructive comments. We believe that the clarification of the presentation of the revised manuscript has been improved.

Response to Reviewer #2:

Overall, this is a good paper which quantifies the uneven urbanization from individual city to global scale. The authors analyzed urbanization characteristics in terms of urban expansion, population growth, built-up area greening and its beneficiaries, and found these characteristics show strong economic-related or spatial patterns. The paper is well-written, clearly organized, and the analysis is well conducted. Hence, I recommend it for publication after addressing the comments suggested below that authors may consider for improving their manuscript.

Response: We are grateful for the reviewer's positive and encouraging comments and invaluable suggestions. Accordingly, we have revised the paper thoroughly. We believe that the clarification of the presentation of the paper has been improved.

1. The concept/term of urbanization is ambiguous in this paper. It is sometimes used to refer to 'urban expansion' (e.g., Section 'Rapid urbanization process'), which is the physical growth of built-up areas. While in some other contexts, the term refers to the general process including urban expansion, population growth and urban environment improvement, e.g., in Lines 160-161. So it would better to clarify and unify the meaning of the term 'urbanization'.

Response: Many thanks for the suggestion. In the revision, the section of 'Rapid urbanization process' was renamed as 'Built-up area expansion'. Yes, urbanization refers to built-up area expansion, urban population growth and environment change. The concept of urbanization has been unified in Introduction section (Lines 39-40, i.e., Notably, urbanization refers to urban built-up area (BUA) expansion, population growth, and environmental change.) and Discussion and conclusions section (Lines 198-199, i.e., Using the results presented above, we identified the urbanization characteristics for large cities in terms of the urban expansion, population growth, and urban environmental changes.).

2. The economic data used in the analysis is based on economic classification at the country level. But the economic level in one single country can be very uneven. Take China as an example. Regions along China's eastern coast are far better developed than other parts of China. Therefore, it's not quite appropriate to group cities like Urumqi in the same economic level with cities/urban agglomerations such as Beijing, PRD and YRD.

I understand that maybe using country-level data is the only feasible way for this analysis at the global scale. So if there are no other suitable economic level datasets, this limitation should at least be mentioned in the discussion section.

For the case studies of PRD and YRD, authors could consider using the economic data of these regions alone instead of the number for the whole country, to quantify the relationship between levels of socio-economic and BUA, population growth, urban greening, etc.

Response: Many thanks for the valuable suggestion. Since the paper analyzed the global data, we didn't include the impacts of regional differences of economy in a country on urbanization development in the study. We assume a country adopt the

similar policies in developing urban areas, and the features of urbanization in one country are distinct from them in another country. Moreover, according to UN's economic data and separation of four income classifications, the paper summarized the urbanization features associated with the income classifications.

Following the comments, in the revision, we emphasized the political zones of economic levels. In the section of discussion and conclusions, we added the limitations of the study in using economic data (see the text on Lines 255-261, i.e., However, due to the singularity of the economic data (income levels)²⁰, the interpretation of the results still has certain limitations. For example, with a large economic aggregate and large land area, China has a significant difference in the economic development in the eastern and western regions. In the YRD and PRD in eastern China, the gross domestic product (GDP) per capita in 2018 (see Methods) has reached the level of the high-income countries. Thus, we should be cautious in elucidating the impacts of country income levels on urbanization.)

3. The term 'Large cities' is commonly defined based on population. In this paper, large cities are urban patches larger than 100 km² extracted from satellite images. Could the authors clarify if this definition will omit cities with a large population or high population density but relatively small areas?

Response: To emphasize the features of urban built-up area expansion in association with urban population growth and greening, this paper defined the large cities in terms of urban patches larger than 100 km². Then, this is a limitation of the study that some cities with large population and small built-up area will not be included. In the revision, we clarified the advantages in the section of built-up area expansion (Lines 62-65, i.e., Further, our analysis focused on the urban patches larger than 100 km² (large cities, hereafter) around the world. The advantage of defining large cities in this way is that we were able to include more cities with large BUA areas but relatively smaller populations, such as many low-density cities in the United States and other developed countries.) and the limitation in the section of discussion and conclusions (Lines 261-264, i.e., In addition, in order to provide a global estimate of urbanization, we used 100 km² as a criterion to screen large cities around the world. Consequently, only dozens of cities in Africa and India were selected as our research objects, which may not represent the reality of urbanization process in these countries.).

4. Fig. 1 and Supplementary Fig. 5: It would be better to consider combining BUA and population, i.e., calculating BUA population density, in Fig. 1 and Supplementary Figure 5. In this way, the situation of urban land expansion lagging behind the needs of the growing urban population (overcrowding) in lower-middle-income and low-income countries can be better presented.

Response: Thanks for the constructive suggestion. We add Fig. 1e to represent the ratio (R_{PB}) of the urban population growth rate to the BUA growth rate of the 105 large cities with BUA expansion exceeding 50 km² from 2001 to 2018. The result presented in Fig. 1e shows that in most parts of the world, urban expansion has lagged far behind the population growth since the advent of the new century.

Moreover, we added a new figure (Fig. 2 in the revised manuscript) and the text on lines 110-120 to present the population density of the 841 large cities. From the new

figure, we can obtain more insights into the urbanization development in recent decades. Fig. 2b shows that the average population density of the large cities in the low-income countries is approximately 3.2 times that in the high-income countries. Concerning the built-up area expansion and population growth, we found that some large cities would face severe crowding and possible other city problems (please see Fig. 2c).

5. Figures 1b, 1d, 2b and 3b: I suggest that the authors conduct a statistical test to see if the average BUA, average PopG, average $REVI$ and the average population living in BUA greening at the four economic levels are significantly different.

Response: Thanks for the suggestion. We have regenerated the Fig. 1b, 1d, Fig. 3b, Fig. 4b and Supplementary Fig. 5b. Those new figures (please see the error bars) include the standard errors of the mean of the corresponding parameters for each economic level. In the revised paper, we analyzed the means of the parameters at different economic levels, and the significant differences of the means were given. (Please see the text on Lines 81-83; namely, "From 2001 to 2018, the average BUA expansion in high-income countries was only 12.6 km² per city, which significantly differs from that of the other three groups ($P < 0.01$; purple lines in Fig. 1b).")

6. Lines 130-139: The "fried egg" shape is very interesting. But it would be better to briefly explain what the "fried egg" shape indicates and why this shape matters. For example, what is the difference between the 'yolk' areas and 'egg white' areas in terms of urban density, population density or time when the urban land appears? Why it's shaped like that?

Response: In the revised manuscript, we included the more explanation about "fried egg", "yolk" and "egg white". Please see the text on Lines 143-156 and below:

From the spatial pattern of the greening pixels in 325 large cities ($REVI > 0.1$), we found that many greening areas of large cities have a shape like a "fried egg", where the yolk-shaped area has been turning green significantly ($P < 0.05$); the "egg white" area refers to the area of browning or no significant greening outside the yolk-shaped area. These yolk-shaped BUA greening areas are usually located in the cores of the cities. Normally, the main characteristics of these areas include a high building density and high population density²⁶. In these areas, the majority of the urban construction would have already been completed before 2001, so any newly established parks or green spaces, even the growth of street vegetation, will increase the greenness of the BUA pixels year by year.

7. Section 'Beneficiaries from urban greening': Greening areas in this context are not necessarily greener than other areas. Just as said in Line 189, some areas are "green enough". So besides beneficiaries of urban greening, I suggest analyzing beneficiaries of urban greenness (EVI_{max}) as well, instead of relating urban greenness to BUA in Supplementary Fig. 6.

Response: Many thanks for the suggestion. We added a paragraph in Lines 179-195. In addition, we regenerated Supplementary Fig. 5a-c. The revised manuscript included the population living in BUA with different levels of average urban

greenness (EVI_{city}). (i.e., Lines 189-195: Furthermore, according to the gridded population data in 2015, only 12% (135 million) of the population lives in one quarter of the large cities with the highest EVI_{city} (Q4), and over 71% (157 out of 211) of the large cities in Q4 are located in high-income countries (Supplementary Fig. 5c). In contrast, approximately 69% of population are lives in large cities with a relatively low EVI_{city} in Q1 and Q2 (Supplementary Fig. 5c).

8. Some other minor questions:

1) Line 9: Please check the grammar. 'Global' is an adjective.

Response: Done (Lines 8-9). i.e., However, a perspective to clarify the urbanization characteristics from individual cities to the globe still lacks.

2) Lines 10-11: In the sentence 'we found lower-middle-income and low-income countries respond to the high population pressure and environmental problems', the word 'respond' seems unclear.

Response: We rewrote the abstract. Please see the text on lines 11-14 and below:

On average, large cities in the low-income and lower-middle-income countries have the highest urban population growth, and the BUA expansion in the upper-middle-income countries is more than three times that of the high-income countries.

3) Line 51: Please indicate if 'adjacency' is determined based on 4-connectivity or 8-connectivity.

Response: Done, see the detail in Method section in Lines 289-291. i.e., In this process, only the pixels that were edge-adjacent (4-connectivity) were merged into a united urban patch.

4) Section 'MODIS land cover type product': The classification accuracy of the MODIS land cover product could be mentioned in order to show the reliability of using this data to extract large cities.

Response: Done. See the detail in Lines 285-287. i.e., The spatial resolution of the land cover product was 500 m and had a reported 73.6% overall land cover classification accuracy and relatively high accuracy for urban areas¹⁹.

5) Lines 222-223: It is not clear how NDVI datasets were used in the study. If it is important, it would be better to give a little more explanation. Otherwise, I suggest deleting the sentence 'the NDVI datasets are used for the comparative analyses'.

Response: Done. We deleted the text. Thank you for the suggestion.

6) The word 'frequency' in all of the figures seems a little unsuitable. 'Percentage' may be the better word. It would also be better if the decimals in Table 1 could be presented as percentages.

Response: Done. In the revision, we used 'percentage' to replace 'frequency' in all the

figures and Table 1.

7) Fig. 3b: The title of the purple y-axis should be 'Average population living in GBUA (thousand)'.

I hope the authors will find my comments useful.

Response: Done. We regenerated Fig. 4b and use 'Average Beneficiaries' for the title. We much appreciated the reviewer's constructive and valuable comments and suggestions.

REVIEWER COMMENTS

Reviewer #1 (Remarks to the Author):

I have reviewed "Dramatic uneven urbanization development of global large cities in recent decades" for a second time. My first review of the paper was very critical, and I concluded that the paper was not publishable. I have carefully given the revised paper a second look at the explicit request of the editors. The paper does address a very important topic (recent urban growth and greening), performed on a comprehensive assessment of all "large" cities throughout the world, and the results are interesting and important. The methods appear to be rigorous (but see major concern about scale of greenness trend used), and the results are interesting. So, I do think the paper has value. While the revisions have improved the paper, there are still significant problems with the paper and I do not think it is publishable in its current form. Notably, there are a number of items I brought up in the original version of the paper that were not appropriately addressed. Specifically, there remains significant bias in the interpretation of the results, where the authors' conclusions go beyond what the data should allow suggesting a subjective bias by the authors in their conclusions. The English remains problematic in many places. I also have one major concern about how greening was measured, which requires clarification and perhaps reanalysis and reinterpretation.

I have tried to be as thorough in my comments here as possible, because it is important that if this paper is to be published, that the interpretation is accurate and the conclusions are appropriate.

1) Concerns with methodology used to quantify urban greening -

The scale at which urban greening was measured is unclear and potentially problematic. Was urban greening measured over the entire extent of the urban area of each city, and readjusted as cities grew? This would be the best approach because it would properly green over the entire urbanized area, without confound from conversion of forest, arid, or rural areas to urban development, which would downwardly bias greening trends.

Alternatively, was a fixed area taken that may or may not contain the entirety of each city. The following text on lines 311-312 suggest it was the latter, but it is still very much unclear from this text:

"the spatial resolution of the MOD13A1 data is 500 m; thus, we wrote a python program to calculate the Mann-Kendall trend for each pixel in a 2,400 by 2,400 km grid"

Surely this same grid was not applied to every city since 2400x2400 km would not include every city? So was a new grid applied to each city, and was it really 2400km x 2400km? Note, the distance from Beijing to Hong Kong isn't even 2000km. How was this grid then restricted to the limits of each city and adjusted through time for their growth? Overall, I am very confused about the scale used to measure urban greening and this scale will have a huge influence on the results obtained, and thus the conclusions reached. For example, if the cities were smaller than the grid used then the pixels would also be picking up rural areas, which we would not expect to significantly green over time, or may decrease in greenness if converted to urban development during 2001-2018. As mentioned above, I would think that the best approach would be to scale the sampling area to each city, and have that scaling change through time to match the growth of city size.

In conclusion, this method needs clarification because it is key to one of the most important questions and conclusions from the paper.

2) Lines 231-232: Discussion and results relating to "green enough" urbanization- I appreciate that "green enough" is in quotations, but this term is misleading and should not be used. It isn't

clear how much green is enough for maintaining sustainable ecosystems and healthy human populations. Existing data suggests that the benefits of increased greening have no clear plateau within the range of urban greenness on both ecosystem health (pick your favorite urban ecology book) and health benefits to humans (e.g. Kardan et al 2015 Sci Reports, Engemann et al 2019 PNAS, the list of possibly refs is long), so this wording and analysis is both inaccurate and unsubstantiated in the message it sends.

3) Biased interpretation: Throughout the paper the authors interpret a significant trend in greening as "improvement" to the environment and as a "benefit" to people. The authors have quantified urban change and urban greening, but they have not explicitly measured any metric of improvement to the environment or the direct effects of urban greening on human health. There are such studies, but not on the scale of analysis performed. Conclusions about "improvement" and "benefits" would need direct measurement of environmental quality (e.g. water quality, air pollution, etc.) and human health (e.g. cardiovascular disease, psychological disorders, etc.). It is important the authors' interpretation and conclusions be limited to their data, and to leave any subjective interpretation to a wider discussion, where they can openly and explicitly speculate about how their results relate to ecosystem and environmental quality and benefits for humans. For example, it would be totally acceptable to describe their results, and then late in the discussion to say that access to nature and the amount of urban green space are likely to have multiple benefits on the environment and for human health/well-being, including cardiovascular condition, mental health and reduced crime and violence. In conclusion, the authors need to be MUCH more cautious in their interpretation and to make it clear to readers when they are speculating.

Here are specific instances of the authors' claims that go beyond their data or that show a bias of interpretation:

- Line 14: "urban environment has improved in 325 large cities"
- Line 32-33: "a pleasant urban environment" is vague and subjective.
- Line 46: "Has the urban environment improved"? The authors need to restate this question – they measured whether urban environments have undergone greening; they have not directly measured "improvement". For example, "greening" in Phoenix is not beneficial to most native arid flora and fauna. Similarly greening achieved by the planting of exotic trees may not benefit native biodiversity, while it may have positive effects on air quality and human well-being. Thus, the effects of greening are nuanced and dependent on one's perspective.
- Line 116-117: "resulting in various urban problems, such as overcrowding or emergence of urban slums." - The authors are assuming "urban problems" without measuring them. The citation given is related to Bangladesh, not the cities being discussed. The authors should not make conclusions that go beyond their data. They can speculate in the discussion, but such speculations should be clearly identified as such, not presented as fact as is done here by the authors.
- Line 119: "Fortunately", again this wording shows a biased interpretation of the data not directly supported by the data
- Line 229: "gratifying to note ...", the word "gratifying" suggests the authors are proud about something, again showing a biased and subjective interpretation of the importance of their results.
- Line 249-252: "As the largest developing country and the second largest economy, China has improved the urban environment dramatically, and the experience of urbanization in China can be a beneficial reference for the other developing countries" - This is not an objective interpretation of the results, but instead biased viewpoint that reads more like propaganda than actual science. The authors have not directly measured improvement to environments or benefits to humans; they have measured greening, with the assumption that this equates to these two things. The authors need to be more careful in their interpretation and tone.

4) Although the authors indicate they have corrected all of the English and used a professional editing service, there are still numerous problems with grammar, word useage and vague or unclear writing. Here are the specific issues I identified:

- Title: The title is grammatically incorrect and doesn't make sense as written. Please consider the

following: "Dramatic uneven urban development of large cities throughout the world in recent decades"

- Line 8: revise to read: "The world has experienced ..." OR "The world has been experiencing ..."
- Lines 8-9: "The globe still lacks" is grammatically incorrect. Consider reword the sentence to read: "However, we still lack information about the characteristics of urbanization from individual cities around the world".
- Line 24: revise to read: "of the United Nations"
- Line 27: What is meant by: "whether the future of urban is flourishing ..."? This is vague and unclear.
- Line 29-30: "urbanization with development occurs when urban expansion is adaptable to the population growth". This is vague and unclear. Furthermore, urbanization by definition includes development, so this doesn't make sense (also applies to line 33)
- Line 116: revise to read: "had limited urban expansion, ..."
- Line 133: "... Revi larger than 0.1." - As written it is not clear to readers what the ratio means in any biological sense. What does a value >0.1 correspond to in concrete terms? Can the authors relate this to a % increase in greening per unit area?
- Line 144-146, re: Fried egg analogy: While this is a nice analogy, can the authors relate this to a location within the city? Is the "yolk" the downtown area and the "egg white" the suburbs? The yolk is defined further down, but it would be helpful if that was given here. The eggs white is never defined with respect to location in the city
- Line 185-186: The point being made here is unclear. Do the authors mean to say that these cities are naturally arid, and are naturally less green because of this? Please be more clear and explicit.
- Line 189-193: These two sentences are poorly written and have many grammatical errors. They need to be completely and carefully rewritten
- Line 235: "that on the way to achieve". Please correct the grammar, I am not sure what the authors are trying to say.
- Line 263-264: the word "objects" is an odd choice here, and the last clause should have the word "the" before "urbanization" ("... the urbanization process...")
- Figure legends: Please define all acronyms used in the figures (e.g. PRD, YRD) in the figure legend, so figures can stand alone.

Other minor comments:

- 5) Lines 74-75: Quote income amounts parenthetically here.
- 6) Line 106: This measure doesn't account for increased vertical growth through the construction of highrise apartments and condominiums, seen in many large cities throughout the world. Some cities are intensifying as opposed to growing outwards

Reviewer #2 (Remarks to the Author):

Thank you for the careful attention in addressing the comments I provided. I think all of them make the paper stronger. However, there are still a few suggestions I would like to give as below.

1. 'Urbanization development' in the title and the text (Lines 200 and 272) is not an accurate term. It should either be termed as 'urbanization' or 'urban development'. I suggest using 'urbanization'.
2. I think the three research questions are so big that it's almost impossible to address any of them in one single paper. Please be more explicit and rewrite them to match what is actually solved.
3. Line 29: The context 'in countries with economic growth and development' is too broad. Lots of countries are experiencing economic growth and development, but urbanization with development is not common in many of them.
4. Lines 39-40: The definition of 'urbanization' is incorrect as urbanization covers more than these three aspects. I would suggest not to define 'urbanization' but focus on clarifying the three perspectives this paper aims to characterize.

5. Line 203: Please clarify what the first feature is at the start of this paragraph instead of stating how this feature was obtained. Similar to this issue, in Line 223, the second feature also needs to be described accurately.

6. Lines 113-115: I don't quite understand why only six cities are mentioned here, because all the dots in Fig. 2c represent cities with population growth more than 2 million. Besides, according to Fig. 2a, among these six cities, Cairo and Dhaka experienced massive urban expansion (> 200 km²).

My remaining concerns are mainly grammatical and wording-related.

1. The word 'the' is used too many times where it's not needed, making the text not very smooth to read. For example, in Line 115, 'the' in 'the limited urban expansion' can be deleted.

2. The use of past tense and present tense should be double-checked. For example, in Lines 285-287, 'was' and 'had' should be replaced with 'is' and 'has'.

3. Some statements are not concise enough, e.g., in Lines 59-60, the sentence 'Except for...'. In Line 202, 'urbanization in recent decades' would be enough. Another example is Lines 194-195, where the sentence could be rewritten in more authentic English.

4. Line 8: should use a positive voice.

5. Line 16: 'benefited' should be followed by 'from...'.

6. Line 27: should use a noun after 'the future of' instead of the adjective 'urban'.

7. Line 193: use 'live' instead of 'are lives'.

8. Line 213: The word 'asymmetric' does not fit in this context.

9. Lines 234-235: should be 'but the ratio of the BUA greening from 2001 to 2018 is also the smallest'.

10. Line 256: 'Singularity' does not seem to be the right word. Could use 'scarcity' instead.

11. Line 268: should read 'achieving'.

12. The s.e.m (standard error of the mean) should be capitalized as S.E.M (Standard Error of The Mean).

Response to Reviewers

Response to Reviewer #1:

I have reviewed “Dramatic uneven urbanization development of global large cities in recent decades” for a second time. My first review of the paper was very critical, and I concluded that the paper was not publishable. I have carefully given the revised paper a second look at the explicit request of the editors. The paper does address a very important topic (recent urban growth and greening), performed on a comprehensive assessment of all “large” cities throughout the world, and the results are interesting and important. The methods appear to be rigorous (but see major concern about scale of greenness trend used), and the results are interesting. So, I do think the paper has value. While the revisions have improved the paper, there are still significant problems with the paper and I do not think it is publishable in its current form. Notably, there are a number of items I brought up in the original version of the paper that were not appropriately addressed. Specifically, there remains significant bias in the interpretation of the results, where the authors’ conclusions go beyond what the data should allow suggesting a subjective bias by the authors in their conclusions. The English remains problematic in many places. I also have one major concern about how greening was measured, which requires clarification and perhaps reanalysis and reinterpretation.

I have tried to be as thorough in my comments here as possible, because it is important that if this paper is to be published, that the interpretation is accurate and the conclusions are appropriate.

Response: We much appreciated your further valuable comments. According to these comments, we have revised the manuscript thoroughly again. The following gives the detailed responses to these comments one-by-one.

1) Concerns with methodology used to quantify urban greening -

The scale at which urban greening was measured is unclear and potentially problematic. Was urban greening measured over the entire extent of the urban area of each city, and readjusted as cities grew? This would be the best approach because it would properly measure greening over the entire urbanized area, without confound from conversion of forest, arid, or rural areas to urban development, which would downwardly bias greening trends.

Alternatively, was a fixed area taken that may or may not contain the entirety of each city. The following text on lines 311-312 suggest it was the latter, but it is still very much unclear from this text: “the spatial resolution of the MOD13A1 data is 500 m; thus, we wrote a python program to calculate the Mann-Kendall trend for each pixel in a 2,400 by 2,400 km grid”.

Surely this same grid was not applied to every city since 2400x2400 km would not include every city? So was a new grid applied to each city, and was it really 2400km x 2400km? Note, the distance from Beijing to Hong Kong isn’t even 2000km. How was this grid then restricted to the limits of each city and adjusted through time for their growth? Overall, I am very confused about the scale used to measure urban greening and this scale will have a huge influence on the results obtained, and thus the

conclusions reached. For example, if the cities were smaller than the grid used then the pixels would also be picking up rural areas, which we would not expect to significantly green over time, or may decrease in greenness if converted to urban development during 2001-2018. As mentioned above, I would think that the best approach would be to scale the sampling area to each city, and have that scaling change through time to match the growth of city size.

In conclusion, this method needs clarification because it is key to one of the most important questions and conclusions from the paper.

Response: Many thanks for the critical comment. Due to the typo in the previous manuscript, the description of calculating EVI trend might mislead an understanding of the method. In fact, the spatial resolution of both Land Cover Product and EVI raster file used in this study is 500 m, and each raster file has 2400 by 2400 rows/columns (NOT 2400 by 2400 km grid). Taking Land Cover Data in 2018 as an example, for the world, there are 212 data tiles that can be used to detect the existence of BUA data.

In this study, we extracted urban patch BUA data larger than 100 km² from the Land Cover Product in 2018. Then, we calculated the trend of annual maximum EVI (E_{\max}) in 2001-2018 for each pixel within an urban extent (BUA in 2018). In the areas transformed from farmlands or forests into urban land from 2001 and 2018, the greenness of these areas would decrease. However, for the original urban areas, if the new parks or green spaces were developed, the greenness of these areas would increase. In this study, we calculated the E_{\max} trend of the pixels in all the large cities, and then identified those pixels with significant trends in each city. Accordingly, we could obtain greening features of all the large cities. As per the comment, to clarify the method, we revised the description of the main text and Method section.

Please see the main text on lines 127-131. i.e., Using the enhanced vegetation index (EVI)^{23,24} as a greenness indicator and BUA of 2018 as urban extent, we applied the Mann-Kendall method to calculate the trend of the annual maximum EVI (E_{\max}) for each urban pixel of the 841 large cities from 2001 to 2018 (see Methods). Then, we calculated the ratio (R_{EVI}) of the area of the pixels with a significant increasing E_{\max} trend ($P < 0.05$) to the BUA in 2018 of the corresponding large city.

In the Method section, we revised the description on lines 307-314. i.e., In this study, the spatial resolution of the MOD13A1 data is 500 m; thus, we wrote a Python program to calculate the Mann-Kendall trend of E_{\max} from 2001 to 2018 for each pixel in a 2,400 by 2,400 rows/columns raster file. Then, the Geospatial Data Abstraction Library (GDAL) (<https://gdal.org>) was used to mosaic all the raster files (212 tiles) of the trends into a global raster map of the E_{\max} trend. Finally, we extracted the E_{\max} trend for each urban pixel within the urban extent (BUA in 2018) and calculated the ratio (R_{EVI}) of the area of the pixels with a significant increasing E_{\max} trend ($P < 0.05$) to the entire BUA of the corresponding urban extent.

2) Lines 231-232: Discussion and results relating to “green enough” urbanization- I appreciate that "green enough" is in quotations, but this term is misleading and should not be used. It isn't clear how much green is enough for maintaining sustainable ecosystems and healthy human populations. Existing data suggests that the benefits of

increased greening have no clear plateau within the range of urban greenness on both ecosystem health (pick your favorite urban ecology book) and health benefits to humans (e.g. Kardan et al 2015 Sci Reports, Engemann et al 2019 PNAS, the list of possibly refs is long), so this wording and analysis is both inaccurate and unsubstantiated in the message it sends.

Response: Thanks for the acute comment. In the revised manuscript, we used "greener" to replace "green enough". Further, we modified the text on lines 183-184. i.e., In other words, these cities should be "greener" compared with those in developing countries. In Discussion and conclusions, lines 229-231. i.e., In the high-income countries, due to "greener" urbanization (Supplementary Fig. 5a and 5b) before 2001, the greening increase of these cities is not the largest (Fig. 3b).

3) Biased interpretation: Throughout the paper the authors interpret a significant trend in greening as "improvement" to the environment and as a "benefit" to people. The authors have quantified urban change and urban greening, but they have not explicitly measured any metric of improvement to the environment or the direct effects of urban greening on human health. There are such studies, but not on the scale of analysis performed. Conclusions about "improvement" and "benefits" would need direct measurement of environmental quality (e.g. water quality, air pollution, etc.) and human health (e.g. cardiovascular disease, psychological disorders, etc.).

It is important the authors' interpretation and conclusions be limited to their data, and to leave any subjective interpretation to a wider discussion, where they can openly and explicitly speculate about how their results relate to ecosystem and environmental quality and benefits for humans. For example, it would be totally acceptable to describe their results, and then late in the discussion to say that access to nature and the amount of urban green space are likely to have multiple benefits on the environment and for human health/well-being, including cardiovascular condition, mental health and reduced crime and violence. In conclusion, the authors need to be MUCH more cautious in their interpretation and to make it clear to readers when they are speculating.

Response: Thanks for the valuable suggestion. In the new revised manuscript, in the Result section, we used "population in greening BUA" to replace "beneficiaries". We described population living in BUA with significant greening trend in Result section, and all the benefits of urban greenness increase are discussed in Discussion section.

See the text on lines 164-167. i.e., there are approximately 1.16 billion urban residents in the 841 large cities, and 192 million urban population are living in the BUA pixels with significant greening trends (Fig. 4a). The average population living in BUA with significant greening trend in the upper-middle-income countries is the largest.....

See the text in Discussion section on lines 236-239. i.e., In addition, as the world's largest source of emissions³⁰, large cities with a significant greening trend may also benefit the health of local urban residents^{31,32}, and, to a certain extent, mitigate the impact of global climate change³³.

Here are specific instances of the authors' claims that go beyond their data or that show a bias of interpretation:

- Line 14: “urban environment has improved in 325 large cities”

Response: To avoid the biased interpretation, we used “urban greenness has increased” to replace the “improved urban environment”. Accordingly, we rewrote the text in the Abstract. **lines 14-15. i.e.**, Globally, the urban greenness has increased by more than 10 percent of the urban BUA in 325 large cities with a significant greening trend ($P < 0.05$).

- Line 32-33: “a pleasant urban environment” is vague and subjective.

Response: We used “green space or parks” (**lines 31-32**) to replace “a pleasant urban environment”.

- Line 46: “Has the urban environment improved”? The authors need to restate this question – they measured whether urban environments have undergone greening; they have not directly measured “improvement”. For example, “greening” in Phoenix is not beneficial to most native arid flora and fauna. Similarly greening achieved by the planting of exotic trees may not benefit native biodiversity, while it may have positive effects on air quality and human well-being. Thus, the effects of greening are nuanced and dependent on one’s perspective.

Response: Thanks for the comments. Concerning the focus of the study, in the revision, we just included two questions. See the text on **lines 41-43. i.e.**, (i) How have global cities grown in recent decades in terms of urban expansion, population growth and urban environmental change? (ii) What are the relationships between the urbanization features and the economic levels?

-Line 116-117: “resulting in various urban problems, such as overcrowding or emergence of urban slums.” - The authors are assuming "urban problems" without measuring them. The citation given is related to Bangladesh, not the cities being discussed. The authors should not make conclusions that go beyond their data. They can speculate in the discussion, but such speculations should be clearly identified as such, not presented as fact as is done here by the authors.

Response: Done. We deleted the related text in Results section and keep the speculations with the word “could” in Discussion. (**lines 217-219. i.e.**, Hence, the cities with a rapid population growth and relatively slow BUA expansion in the lower-middle-income and low-income countries could cause serious urban problems, such as slums and crowding^{15,28})

- Line 119: “Fortunately”, again this wording shows a biased interpretation of the data not directly supported by the data

Response: Done. See the text on **lines 118-119. i.e.**, Due to the rapid urban expansion, the population densities of these four cities were relatively low

- Line 229: “gratifying to note ...”, the word “gratifying” suggests the authors are proud about something, again showing a biased and subjective interpretation of the importance of their results.

Response: Done. We deleted the text of “It is gratifying to note that” (see the text on

line 226).

- Line 249-252: “As the largest developing country and the second largest economy, China has improved the urban environment dramatically, and the experience of urbanization in China can be a beneficial reference for the other developing countries”
- This is not an objective interpretation of the results, but instead biased viewpoint that reads more like propaganda than actual science. The authors have not directly measured improvement to environments or benefits to humans; they have measured greening, with the assumption that this equates to these two things. The authors need to be more careful in their interpretation and tone.

Response: Many thanks for the valuable comments. We rewrote the text (see the text on lines 249-251). i.e., The greening of cities implies more urban parks or green spaces, which are likely to have multiple benefits to the environment and human health/well-being, including cardiovascular condition^{1, 2}, mental health³ and reduced crime and violence⁴.

4) Although the authors indicate they have corrected all of the English and used a professional editing service, there are still numerous problems with grammar, word usage and vague or unclear writing. Here are the specific issues I identified:

- Title: The title is grammatically incorrect and doesn't make sense as written. Please consider the following: “Dramatic uneven urban development of large cities throughout the world in recent decades”

Response: Many thanks for the kind suggestion. Accordingly, we changed the title as “Dramatic uneven urbanization of large cities throughout the world in recent decades”.

- Line 8: revise to read: “The world has experienced ...” OR “The world has been experiencing ...”

Response: Done, see the text on line 8. i.e., “The world has experienced dramatic urbanization in recent decades.”

- Lines 8-9: “The globe still lacks” is grammatically incorrect. Consider reword the sentence to read: “However, we still lack information about the characteristics of urbanization from individual cities around the world”.

Response: Many thanks. Done, see the text on lines 8-9. i.e., “However, we still lack information about the characteristics of urbanization from individual cities around the world.”

- Line 24: revise to read: “of the United Nations”

Response: Done. see the text on line 24. i.e., “According to the latest report of the United Nations (UN).....”

- Line 27: What is meant by: “whether the future of urban is flourishing ...”? This is vague and unclear.

Response: We rewrote the text. See the text on lines 26-27. i.e., “Undoubtedly, urban sustainable development of city is highly related to the future of humanity”

- Line 29-30: “urbanization with development occurs when urban expansion is adaptable to the population growth”. This is vague and unclear. Furthermore, urbanization by definition includes development, so this doesn’t make sense (also applies to line 33)

Response: We rewrote the text on lines 29-30. i.e., Usually, in developed countries, the urban expansion is adaptable to the population growth. We also rewrote the text on lines 32-33. i.e., On the contrary, for many developing countries, the national economic growth and development are inadequate to meet the needs of a growing urban population.

- Line 116: revise to read: “had limited urban expansion, ...”

Response: Done.

- Line 133: “... Revi larger than 0.1.” - As written it is not clear to readers what the ratio means in any biological sense. What does a value >0.1 correspond to in concrete terms? Can the authors relate this to a % increase in greening per unit area?

Response: To clarify the meaning of R_{EVI} , we rewrote the text on lines 132-134. i.e., There are 325 large cities where more than 10 percent of the urban pixels (i.e., $R_{EVI} > 0.1$) were significantly greening from 2001 to 2018 (red points in Fig. 3a).

- Line 144-146, re: Fried egg analogy: While this is a nice analogy, can the authors relate this to a location within the city? Is the "yolk" the downtown area and the "egg white" the suburbs? The yolk is defined further down, but it would be helpful if that was given here. The egg white is never defined with respect to location in the city

Response: Many thanks. Accordingly, we rewrote the text on lines 144-148. i.e., we found that many greening areas of large cities have a shape like a “fried egg”, where the yolk-shaped downtown area has been turning green significantly ($P < 0.05$); the “egg white” area refers to the area of browning or no significant greening outside the yolk-shaped area, and these areas are usually the suburbs.

- Line 185-186: The point being made here is unclear. Do the authors mean to say that these cities are naturally arid, and are naturally less green because of this? Please be more clear and explicit.

Response: Yes. To clarify the point, we rewrote the text on lines 186-188. i.e., Notably, due to the semi-arid or arid climate, the urban greenness of large cities located in the southwest of the United States and Australia are usually low (red points of Q1 in Supplementary Fig. 5a).

- Line 189-193: These two sentences are poorly written and have many grammatical errors. They need to be completely and carefully rewritten

Response: We rewrote the text on lines 189-195. i.e., The average EVI_{city} in the low-

income countries is 0.28. Furthermore, according to the gridded population data in 2015, for Q4, there were only 12% (135 million) of the total population in the large cities; moreover, over 71% (157 out of 211) of these large cities in Q4 were located in high-income countries (Supplementary Fig. 5c). By contrast, for Q1 and Q2, there were approximately 69% of the total population of the 841 large cities (Supplementary Fig. 5c).

- Line 235: “that on the way to achieve”. Please correct the grammar, I am not sure what the authors are trying to say.

Response: We rewrote the text on lines 234. i.e., These results indicate that on the way to equitable and sustainable urban development

- Line 263-264: the word “objects” is an odd choice here, and the last clause should have the word “the” before “urbanization” (“... the urbanization process...”)

Response: We rewrote the text on lines 261-262. i.e., Consequently, in this study, only dozens of the cities in Africa and India were selected, which may not represent the reality of the urbanization process in those countries.

- Figure legends: Please define all acronyms used in the figures (e.g. PRD, YRD) in the figure legend, so figures can stand alone.

Response: Done. We have defined all acronyms in figures to ensure the all figures can stand alone.

Other minor comments:

5) Lines 74-75: Quote income amounts parenthetically here.

Response: Done. See the text on lines 70-74. i.e., the 841 large cities were divided into four groups: high-income (H, gross national income (GNI) more than \$12,375 per capita), upper-middle-income (UM, GNI between \$3,996 and \$12,375 per capita), lower-middle-income (LM, GNI between \$1,026 and \$3,995 per capita), and low-income (L, GNI less than \$1,025 per capita)

6) Line 106: This measure doesn't account for increased vertical growth through the construction of highrise apartments and condominiums, seen in many large cities throughout the world. Some cities are intensifying as opposed to growing outwards

Response: Done. See the text on lines 105-107. i.e., This result would indicate that in most parts of the world, without considering the vertical growth through the construction of high-rise apartments and condominiums, urban expansion has lagged far behind the population growth since the advent of the new century.

Response to Reviewer #2:

Thank you for the careful attention in addressing the comments I provided. I think all of them make the paper stronger. However, there are still a few suggestions I would like to give as below.

1. 'Urbanization development' in the title and the text (Lines 200 and 272) is not an accurate term. It should either be termed as 'urbanization' or 'urban development'. I suggest using 'urbanization'.

Response: Thanks for the suggestion. Accordingly, we have revised the title and the related text.

2. I think the three research questions are so big that it's almost impossible to address any of them in one single paper. Please be more explicit and rewrite them to match what is actually solved.

Response: Thanks for the valuable comments. In the revision, we narrowed the research questions, and just focused on two questions. Please the text on lines 41-43. i.e., (i) How have global cities grown in recent decades in terms of urban expansion, population growth and urban environmental change? (ii) What are the relationships between the urbanization features and the economic levels?

3. Line 29: The context 'in countries with economic growth and development' is too broad. Lots of countries are experiencing economic growth and development, but urbanization with development is not common in many of them.

Response: We rewrote the text on lines 29-30. i.e., Usually, in developed countries, the urban expansion is adaptable to the population growth.

4. Lines 39-40: The definition of 'urbanization' is incorrect as urbanization covers more than these three aspects. I would suggest not to define 'urbanization' but focus on clarifying the three perspectives this paper aims to characterize.

Response: Many thanks for your kind suggestion. We have deleted the context of definition. In Discussion and conclusions, we clarified those three perspectives. See the text on lines 198-199. i.e., we identified the urbanization characteristics for large cities in terms of the urban expansion, population growth, and greening BUA changes.

5. Line 203: Please clarify what the first feature is at the start of this paragraph instead of stating how this feature was obtained. Similar to this issue, in Line 223, the second feature also needs to be described accurately.

Response: Done. Please see the text on line 203. i.e., The first is the unevenness between BUA expansion and urban population growth.

Also, please see the text on lines 222-223. i.e., The second feature is about the unevenness between the rapid urban population growth and slow urban environmental improvement.

6. Lines 113-115: I don't quite understand why only six cities are mentioned here, because all the dots in Fig. 2c represent cities with population growth more than 2 million. Besides, according to Fig. 2a, among these six cities, Cairo and Dhaka experienced massive urban expansion ($> 200 \text{ km}^2$).

Response: These cities are the top six large cities with the largest population density. We rewrote the text on lines 113-116. i.e., Among these large cities, Dhaka, Kathmandu, Manila, Karachi, Istanbul, and Cairo are the top six large cities with the highest population density (Fig. 2c). These cities have experienced rapid population growth (> 2 million) from 2000 to 2015 (Fig. 2c), and limited urban expansion from 2001 to 2018 (less than 200 km^2 , Fig. 2c).

My remaining concerns are mainly grammatical and wording-related.

1. The word 'the' is used too many times where it's not needed, making the text not very smooth to read. For example, in Line 115, 'the' in 'the limited urban expansion' can be deleted.

Response: Done. We have removed "the". Thank you. (see the text on line 116)

2. The use of past tense and present tense should be double-checked. For example, in Lines 285-287, 'was' and 'had' should be replaced with 'is' and 'has'.

Response: Done. (see the text on lines 282-283)

3. Some statements are not concise enough, e.g., in Lines 59-60, the sentence 'Except for...'. In Line 202, 'urbanization in recent decades' would be enough. Another example is Lines 194-195, where the sentence could be rewritten in more authentic English.

Response: Done.

See the text on lines 56-57. i.e., Except for the United States, the other nine countries are developing and emerging countries.

See the text on lines 201-202. i.e., Explicitly, we highlight below three urbanization features obtained in this study.

Regarding the sentence on lines 194-195, in order to be more concise, we deleted the sentence.

4. Line 8: should use a positive voice.

Response: Done. See the text on Line 8. i.e., The world has experienced dramatic urbanization in recent decades.

5. Line 16: 'benefited' should be followed by 'from...'.

Response: We revised the sentence. See the text on Line 15-17. i.e., Particularly, China accounts for 32% of the greening BUA of the 841 large cities, where about 108 million people have been living.

6. Line 27: should use a noun after 'the future of' instead of the adjective 'urban'.

Response: We revised the sentence. See the text on Line 26-27. i.e., Undoubtedly, urban sustainable development is highly related to the future of humanity

7. Line 193: use 'live' instead of 'are lives'.

Response: We revised the sentence. See the text on Line 194-195. i.e., By contrast, for Q1 and Q2, there were approximately 69% of the total population of the 841 large cities (Supplementary Fig. 5c).

8. Line 213: The word 'asymmetric' does not fit in this context.

Response: Done. See the text on Line 212. i.e., Moreover, due to the different growth rates of urban population and urban land.

9. Lines 234-235: should be 'but the ratio of the BUA greening from 2001 to 2018 is also the smallest'.

Response: Done. See the text on Line 233. i.e., but the ratio of the greening BUA from 2001 to 2018 is also the smallest

10. Line 256: 'Singularity' does not seem to be the right word. Could use 'scarcity' instead.

Response: Done. See the text on Line 254. i.e., However, due to the scarcity of the economic data (income levels).....

11. Line 268: should read 'achieving'.

Response: Done. See the text on Line 266. i.e.,is essential to achieving the UN's Sustainable Development Goals.....

12. The s.e.m (standard error of the mean) should be capitalized as S.E.M (Standard Error of The Mean).

Response: Thanks for the suggestion. Done.

REVIEWER COMMENTS

Reviewer #2 (Remarks to the Author):

Thanks for the editor's trust and invitation to also take a look at the responses to Reviewer #1's comments, as Reviewer #1 is unable to provide a third review this time. Overall, some concerns of the reviewers have been addressed. However, there still exist a few major issues, mainly with the 'greenness' issue raised by Reviewer #1. The English in the paper also needs another thorough check preferably by native speakers, because many sentences are still with grammar or wording problems, or unclear. Specific comments are given as below.

1. Reviewer #1's concern about the usage of BUA in greenness trend calculation is not fully resolved. In my opinion, compared to BUA in 2018 and BUA of every year, it makes the most sense to use stable BUA (pixels that were built-up areas throughout the whole study time period) as the extent of calculating greenness trend.

λThe reason for not using BUA in 2018:

The paper focuses on the urban greenness, but the greenness of rural vegetation (cropland and natural vegetation) is not urban greenness. So it does not make sense in this paper's context to measure the greenness trend of a pixel that changed from rural vegetation to built-up area (urban expansion).

λThe reason for not using the BUA of every year:

BUA is different each year, so the trend calculation will be impracticable.

If the authors have solid reasons to use BUA in 2018 when measuring the greenness trend, then it would be necessary to clarify in the text the reason for doing so.

2. Regarding Reviewer #1's comment on greenness and 'improvement' to the environment, there remain many sentences in the paper where greenness increase is still interpreted or referred to as 'environmental improvement', e.g., in Lines 223, 229, 246, 275. Please carefully check this issue.

3. Similar to the previous issue, in the research questions (Lines 41-43), 'urban environmental improvement' is different from urban greenness increase and was not measured in the paper. So I suggest rewriting the first research question as 'How have global large cities developed in recent decades in terms of urban expansion, population growth and urban greenness change?'

In Lines 121 and 122, 'Urban environmental change' should also be reworded as 'Urban greenness change'. Please double-check for other places where this issue exists.

4. I suggest using EVImax instead of Emax, as it is a bit confusing.

5. As mentioned by Reviewer #1, REVI is indeed misleading. Could consider renaming this variable as Rgreening.

6. Line 16: According to the first paragraph in the 'Population in greening BUA' section, '108 million people' is obtained from the population data in 2015, so the usage of 'have been living' is inaccurate. Could use 'lived' instead.

7. Lines 14-15: The sentence 'Globally, the urban greenness...' is unclear and its meaning turns out not what you meant to say. Could revise it to read: 'Globally, more than 10 percent of BUAs in 325 large cities indicate significant greening ($P < 0.05$) from 2001 to 2018.'

8. Lines 30-31: 'public services' and 'urban infrastructure and green space or parks' do not preclude each other. Please consider rewording the sentence to read: "The residents are served by good public services and have access to adequate urban infrastructure, such as water and energy supplies, sanitation, education, and green space or parks."

9. Line 44: Delete 'the' in 'to address both the questions'.

10. Lines 171-173: This sentence is too long. Please consider splitting it into two sentences.

11. Line 183: Add 'before 2001' after 'should be greener'.

12. Lines 184-185: Please explain the reason for calculating the average EVIcity in 2018 instead of that in/before 2001. Because according to previous sentences, the point here is to prove large cities in high-income countries are 'greener' before 2001 (the 'initial' stage).

13. Lines 191-192 and 194-195: These two sentences are unclear and confusing. Please consider rewording them to read:

"Furthermore, according to the gridded population data in 2015, only 12% of the large cities' total population lived in cities in Q4."

"By contrast, approximately 69% of the large cities' total population lived in cities of Q1 and Q2. (Supplementary Fig. 5c)."

14. Lines 292-293: Please check grammar.

15. Line 295: Revise 'Urban BUA' to read 'BUA'.

16. Line 296: Should read 'this is the reason why...'

17. Line 309: I don't think it is necessary to mention the image size (2400 by 2400 pixels). Because all the tiles were eventually mosaicked, the size of the images is not very relevant information to the analysis.

18. Please consider using 'over' or 'more than' instead of 'exceeding' in most cases, e.g., in 'excluding 19 large cities exceeding 2000 km²' (Line 294).

Response to Reviewers

Response to Reviewer # 2:

Thanks for the editor's trust and invitation to also take a look at the responses to Reviewer #1's comments, as Reviewer #1 is unable to provide a third review this time. Overall, some concerns of the reviewers have been addressed. However, there still exist a few major issues, mainly with the 'greenness' issue raised by Reviewer #1. The English in the paper also needs another thorough check preferably by native speakers, because many sentences are still with grammar or wording problems, or unclear. Specific comments are given as below.

Response: We much appreciated your further valuable comments. According to these comments, we have revised the manuscript again. The following gives the detailed responses to these comments one-by-one.

1. Reviewer #1's concern about the usage of BUA in greenness trend calculation is not fully resolved. In my opinion, compared to BUA in 2018 and BUA of every year, it makes the most sense to use stable BUA (pixels that were built-up areas throughout the whole study time period) as the extent of calculating greenness trend.

The reason for not using BUA in 2018:

The paper focuses on the urban greenness, but the greenness of rural vegetation (cropland and natural vegetation) is not urban greenness. So it does not make sense in this paper's context to measure the greenness trend of a pixel that changed from rural vegetation to built-up area (urban expansion).

The reason for not using the BUA of every year:

BUA is different each year, so the trend calculation will be impracticable.

If the authors have solid reasons to use BUA in 2018 when measuring the greenness trend, then it would be necessary to clarify in the text the reason for doing so.

Response: Many thanks for the critical comments.

During our study, we have considered which year of the BUA data should be used to explore urban greening features. We agreed that in calculating the change of BUA greenness, it makes the most sense to use stable BUA (namely, BUA in 2001). However, we found that if only stable BUA was used, the greenness change of the urban pixels, which were developed in recent years, will not be taken into account. For example, if a pixel changed to urban area since 2002 or after, the greenness change in the pixel will not be calculated if only stable BUA is used. Therefore, we used the BUA dataset in 2018, which is the latest data available, to explore the greening features. In order to have a comprehensive and systematic analysis of the global urban greenness changes, we thought it is reasonable to use BUA in 2018 to detect the urban greening change.

Also, we have considered the possible influence from the lands from rural vegetation (cropland and natural vegetation) to BUA. Therefore, in our study, we calculated the BUA where the greenness has a significant increasing trend ($P < 0.05$).

Using this method, we could exclude those pixels that changed from rural vegetation to BUA (urban expansion). In fact, from the paper, we found that most of the greening BUA pixels we obtained are located in the stable urban pixels that were built-up areas throughout the whole study time period (please see the text on Lines 145-161 about the greenings areas with a shape like a “fried egg”).

BUA is different each year. Nevertheless, the difference is related BUA expansion. In the study, we used the annual maximum EVI (EVI_{max}) for each urban pixel of the 841 large cities from 2001 to 2018 to compute the greening trend. If the expansion was obtained from the rural vegetation, the method can exclude those newly expanded BUA. If the newly expanded BUA was originally located in desert and arid areas, such as Las Vegas and Dubai, the method could detect those expanded BUA urban pixels with significant greening trend.

To clarify the reason of using the BUA in 2018, in the revised manuscript, please see the main text on Lines 130-132. *i.e.*, Using this method, we evaluated urban greening change since 2001. Most of the pixels with significant greening trends were located in stable BUAs (*i.e.*, pixels that were BUAs throughout the study period, 2001-2018; see Methods).

Also, please see the text about Methods on Lines 310-316. *i.e.*, For a comprehensive and systematic analysis of global urban greenness change, we used the BUA in 2018 to detect urban greening change. Then, if a pixel changed to an urban area in 2002 or later, the greenness change in the pixel was calculated. Notably, if the expended BUA was developed from cropland and natural vegetation land, the greenness of the expended BUA would possibly decrease. In fact, we calculated the BUA where greenness showed a significant increasing trend ($P < 0.05$). Using this method, we could exclude those pixels that had changed from rural vegetation to BUA (urban expansion).

2. Regarding Reviewer #1’s comment on greenness and ‘improvement’ to the environment, there remain many sentences in the paper where greenness increase is still interpreted or referred to as ‘environmental improvement’, *e.g.*, in Lines 223, 229, 246, 275. Please carefully check this issue.

Response: We removed the term of “environmental improvement” in the main text. Please see the text on:

Lines 224-225. *i.e.*, We used the annual maximum greenness (EVI_{max}) to represent the best state of urban greening.

Lines 228-230. *i.e.*, However, with the economic development of the upper-middle-income countries in recent decades, urban greening has increased significantly,.....

Lines 245-246. *i.e.*, Among the 325 large cities in the world with significant urban greening increase ($R_{greening} > 0.1$),.....

Lines 273-274. *i.e.*,thereby achieving a virtuous circle among urban expansion, population growth, and urban greening changes.

3. Similar to the previous issue, in the research questions (Lines 41-43), ‘urban environmental improvement’ is different from urban greenness increase and was not measured in the paper. So I suggest rewriting the first research question as ‘How have global large cities developed in recent decades in terms of urban expansion, population growth and urban greenness change?’

In Lines 121 and 122, ‘Urban environmental change’ should also be reworded as ‘Urban greenness change’. Please double-check for other places where this issue exists.

Response: Thank you for the comments and suggestions. Done.

Please see the main text on Lines 40-42. i.e., (i) How have global large cities developed in recent decades in terms of urban expansion, population growth, and urban greenness change?

Please see the main text on Line 121. i.e., ... global urban greenness change ...

4. I suggest using EVI_{max} instead of E_{max} , as it is a bit confusing.

Response: Many thanks. We have replaced all ‘ E_{max} ’ to ‘ EVI_{max} ’ in the main text, figures and tables.

5. As mentioned by Reviewer #1, REVI is indeed misleading. Could consider renaming this variable as $R_{greening}$.

Response: Many thanks. Done. In this revised version, we use ‘ $R_{greening}$ ’ instead of ‘ $REVI$ ’ in the main text, figures and tables.

6. Line 16: According to the first paragraph in the ‘Population in greening BUA’ section, ‘108 million people’ is obtained from the population data in 2015, so the usage of ‘have been living’ is inaccurate. Could use ‘lived’ instead.

Response: Done.

7. Lines 14-15: The sentence ‘Globally, the urban greenness...’ is unclear and its meaning turns out not what you meant to say. Could revise it to read: ‘Globally, more than 10 percent of BUAs in 325 large cities indicate significant greening ($P < 0.05$) from 2001 to 2018.’

Response: Many thanks. Done.

8. Lines 30-31: ‘public services’ and ‘urban infrastructure and green space or parks’ do not preclude each other. Please consider rewording the sentence to read: ”The

residents are served by good public services and have access to adequate urban infrastructure, such as water and energy supplies, sanitation, education, and green space or parks.”

Response: Many thanks. Done.

9. Line 44: Delete ‘the’ in ‘to address both the questions’.

Response: Done.

10. Lines 171-173: This sentence is too long. Please consider splitting it into two sentences.

Response: Done.

Please see the main text on Lines 172-174. i.e., As the most populous country, China has 150 large cities, and there are about 108 million urban population living in greening BUAs. This accounts for 56.2% of the total population in greening BUAs among the 841 large cities (Fig. 4c).

11. Line 183: Add ‘before 2001’ after ‘should be greener’.

Response: Done. See the text on Line 184.

12. Lines 184-185: Please explain the reason for calculating the average EVI_{city} in 2018 instead of that in/before 2001. Because according to previous sentences, the point here is to prove large cities in high-income countries are ‘greener’ before 2001 (the ‘initial’ stage).

Response: May thanks for the comment. Accordingly, we added Supplementary Fig. 5a to present the average greenness of cities in the initial stage in 2001. Notably, since the EVI data in 2018 were used to study greenness trends, we kept the greenness figure in 2018 (Supplementary Fig. 5b) for comparison. Further, we compared the greenness of cities and evaluated the population lived in large cities in the latest stage in 2018 (Supplementary Fig. 5c and 5d).

In the revised manuscript, please see the text on Lines 181-185, i.e., In contrast, in the high-income countries, there were quite possibly already many parks and green spaces in the cities^{4,26,27} before 2001 (Supplementary Fig. 5a). Even though the urban greenness of upper-middle-income countries increased significantly in recent decades, cities in the high-income countries in 2018 were still “greener” compared to those in developing countries.

13. Lines 191-192 and 194-195: These two sentences are unclear and confusing. Please consider rewording them to read: “Furthermore, according to the gridded population data in 2015, only 12% of the large cities' total population lived in cities

in Q4.” “By contrast, approximately 69% of the large cities' total population lived in cities of Q1 and Q2. (Supplementary Fig. 5c).”

Response: Many thanks. Done.

14. Lines 292-293: Please check grammar.

Response: Many thanks. We revised the sentence.

Please see the text on Lines 292-294, i.e., The reason for using 50 km² as the threshold was that when excluding 12 large cities with BUA expansions of more than 250 km², we found that the value of the average urban expansion of the remaining 829 large cities, adding 1 standard deviation to those expansions, was close to 50 km².

15. Line 295: Revise ‘Urban BUA’ to read ‘BUA’.

Response: Done.

We also revised the text on Lines 295-300, i.e., In addition, there were three levels of BUA (Fig. 1a). The BUA of the first level was between 100 and 580 km², the second level was between 580 and 2000 km², and the third level was larger than 2000 km². For BUAs in 2018, more than 19 large cities had areas over 2000 km². Excluding those 19 cities, we found that the value of the average BUA of the remaining 822 large cities, adding 1 standard deviation to those areas, was close to 580 km². This is why we divided the 841 large cities into three different sizes (i.e., the point sizes in Fig. 1).

16. Line 296: Should read ‘this is the reason why...’.

Response: Thank you. Done.

17. Line 309: I don't think it is necessary to mention the image size (2400 by 2400 pixels). Because all the tiles were eventually mosaicked, the size of the images is not very relevant information to the analysis.

Response: Thank you. We have deleted the related text.

18. Please consider using ‘over’ or ‘more than’ instead of ‘exceeding’ in most cases, e.g., in ‘excluding 19 large cities exceeding 2000 km²’ (Line 294).

Response: Following your suggestion, we used ‘over’ instead ‘exceeding’ for most cases. Please see the text on Lines 10, 84, 102, 116, 297, 480 and 485.

REVIEWERS' COMMENTS:

Reviewer #2 (Remarks to the Author):

The author has completed the revision as required. I think the paper is qualified to be published.